# Estimation of Nitrogen Content Based on the Hyperspectral Vegetation Indexes of Interannual and Multi-Temporal in Cotton

**Lulu Ma [1,†], Xiangyu Chen [1,†], Qiang Zhang [1], Jiao Lin [2], Caixia Yin [1], Yiru Ma [1], Qiushuang Yao [1], Lei Feng [3], Ze Zhang [1,\*] and Xin Lv [1]**

1 The Key Laboratory of Oasis Eco-Agriculture, College of Agriculture, Shihezi University, Shihezi 832003, China; malulu@stu.shzu.edu.cn (L.M.); chenxinagyu@stu.shzu.edu.cn (X.C.); 20182312019@stu.shzu.edu.cn (Q.Z.); yincaixia@stu.shzu.edu.cn (C.Y.); mayiru@stu.shzu.edu.cn (Y.M.); 20182012108@stu.shzu.edu.cn (Q.Y.); luxin@shzu.edu.cn (X.L.)
2 College of Agriculture, Tarim University, Alar 843300, China; linjiao@taru.edu.cn
3 College of Biosystems Engineering and Food Science, Zhejiang University, Hangzhou 310058, China; lfeng@zju.edu.cn
\* Correspondence: zhangze1227@shzu.edu.cn
† These authors contributed equally to this work.

**Abstract:** Crop nitrogen is an efficient index for estimating crop yield. Using hyperspectral information to monitor nitrogen in cotton information in real time can help guide cotton cultivation. In this study, we used drip-irrigation cotton in Xinjiang as the research object and employed various nitrogen treatments to explore the correlation between hyperspectral vegetation indexes and leaf nitrogen concentration (LNC) and the canopy nitrogen density (CND) of cotton in different growth periods and interannual. We employed 30 published hyperspectral vegetation indexes obtained through spectral monitoring in 2019 and 2020 to screen for hyperspectral vegetation indexes highly correlated with the nitrogen in cotton indexes. Based on the same group of hyperspectral vegetation indexes, interannual and multi-temporal nitrogen estimation models of cotton were established using three modeling methods: simple multiple linear regression (MLR), partial least-squares regression (PLSR), and support vector regression (SVR). The results showed the following: (1) The correlations between LNC and CND and vegetation index in individual growth periods of cotton were lower than those for the entire growth period. The correlations between hyperspectral vegetation indexes and cotton LNC, CND, leaf area index (LAI), and aboveground biomass (AGB), were significantly different between years and varieties. The relatively stable indexes between vegetation and LNC were TCARI, PRI, CCRI, and SRI-2, and the absolute values of correlation were 0.251~0.387, 0.239~0.422, 0.245~0.387, and 0.357~0.533. In addition, the correlation between CIred-edge and REIlinear and group indicators (CND, AGB, and LAI) was more stable. (2) In the models established by MLR, PLSR, and SVR, the $R^2$ value from the SVR method was higher in the estimation model based on the entire growth period data and LNC and CND. (3) Using the same group of selected hyperspectral vegetation indexes to estimate nitrogen in cotton in different growth stages, the accuracy of the estimation model of canopy nitrogen density (CND) was higher than that of the estimation model for leaf nitrogen concentration. The canopy nitrogen density most stable model was established by MLR at the flowering and boll stages and the full-boll stage with $R^2 = 0.532~0.665$. This study explored the application potential of hyperspectral vegetation indexes to the nitrogen of drip-irrigated cotton, and the results provide a theoretical basis for hyperspectral monitoring for crop nutrients and canopy structure.

**Keywords:** vegetation index; canopy nitrogen density; different growth stage; support vector regression





## 1. Introduction

Cotton is an important global cash crop. During the past eight years, Xinjiang's cotton yield has accounted for 90.27% of China's yield and 21.6% of global cotton production.

Xinjiang's consumption accounts for 67.5% in China and 22.5% worldwide. Nitrogen is a necessary nutrient element for cotton growth. It plays an important role in crop photosynthesis, growth, and development, and is one of the core factors determining crop biomass and yield [1,2]. The leaf is the major organ for crop photosynthesis. Crop leaf nitrogen content (individual) and canopy nitrogen density (population) not only reflect the nitrogen nutrition status and growth characteristics of plants but also significantly affect crop yield and quality. Research has shown that about 50% of the nitrogen contained in leaves can be used after leaf senescence. Therefore, real-time monitoring of crop nitrogen is conducive to the precision measurement of the nutrient content of crops.

Hyperspectral monitoring is considered by researchers to be a promising tool and technology for nondestructive detection of crop nitrogen, mainly due to the strong correlation between nitrogen and other variables (e.g., chlorophyll), and researchers have found characteristic wavelengths for nitrogen estimation in vegetation spectra [3–5]. The spectral index comprises the reflectance of multiple spectral bands obtained by mathematical calculation. The research shows that the model established by vegetation indexes is more stable and is more widely used for the inversion of crop physiological and nutrient information [6]. At present, researchers mostly use vegetation indexes to calculate crop physiological and biochemical indexes [7,8]. The research results of Farrah et al. [9] confirmed that normalized difference vegetation index (NDVI) could be used as an effective method to estimate the leaf nitrogen status of cotton at different growth stages. Wang et al. [10] estimated leaf nitrogen content (n%), canopy nitrogen density (CND), and nitrogen nutrition index (NNI) of winter wheat during the entire growth period by using vegetation indexes. This research showed that the correlations between simple ratio pigment index (SRPI), modified red-edge simple ratio index (mSR705), ratio index-1dB (RI-1dB), Vogelmann red-edge index (VOG), and red-edge position based on linear interpolation method (REPliner) and each nitrogen index were not significantly affected by growth period, and the estimation model $R^2$ for CND was more than 81%. The accuracy of the estimation model was better than the accuracy of N% and NNI, but the model will be saturated when using a single vegetation index to estimate CND. Ciganda et al. [11] showed that the red-edge chlorophyll index CIred-edge was sensitive to the canopy structure. Tarpley et al. [12] analyzed the relationship between nitrogen concentration in cotton leaves and multiple spectral ratio indexes and conducted cluster analysis according to the prediction accuracy and overall accuracy. It was found that the prediction accuracy and overall accuracy were relatively high when using the ratio of red-edge position to the near-infrared band. Li [13] studied the estimation of the winter wheat spectral index in different areas, years, varieties, and growth stages. Their results showed that the simple ratio of reflectivity at 370 nm and 400 nm (R370/R400) displayed the most consistent estimation accuracy in an indoor experiment ($R^2$ = 0.58) and field experiment ($R^2$ = 0.51), indicating that the growth stage had a significant impact on the performance of different vegetation indexes and the selection of a sensitive wavelength for plant nitrogen concentration (PNC) estimation. Studies have shown that when using the spectral index to estimate crop nitrogen, there are clear differences in the applicable spectral index for different crops or different varieties and ecological areas of the same crop [14,15].

In addition, modeling methods such as deep machine learning can achieve better prediction effects than using sensitive spectral features alone or vegetation indexes [16], and these methods can be applied to the monitoring of crop nutrients and growth indicators [17–19]. Yao et al. [20] compared the methods of artificial neural network and support vector machine regression (SVR) and showed that SVR was the preferred method for estimating crop nutrient contents. The authors suggested that an artificial neural network is suitable for the establishment of models with large sample sizes. Wang et al. [15] studied the generalized PLSR (gPLSR) model using the hyperspectral reflectance of leaves, and this method could well retrieve leaf nitrogen concentration (r = 0.85).

In the application of vegetation indexes, most researchers have used the data of the entire growth period to estimate nitrogen. There are few research results for crop nitrogen

hyperspectral monitoring in different growth periods. In the process of cotton planting in Xinjiang, the technology of water and fertilizer integration is used for nitrogen application. Xinjiang cotton needs eight times the amount of water and fertilizer drip irrigation in the whole growth period, so more attention should be paid to the nitrogen nutrition status in different growth periods of cotton. Real-time and accurate estimation of the nutrient status of cotton in each growth period would be more conducive to the accurate management of cotton and the optimization of yield. The present study is based on data for the leaf nitrogen concentration (LNC) and canopy nitrogen density (CND) of two varieties of drip-irrigated cotton at different growth stages from April 2019 to September 2020. Some relatively stable vegetation indexes were selected by using Pearson's correlation analysis of 30 hyperspectral vegetation indexes and two nitrogen indexes (LNC and CND) employed in three modeling approaches: simple multiple linear regression (MLR), partial least-squares regression (PLSR), and support vector regression (SVR). The models were used to explore the potential of estimating the nitrogen nutrition status of cotton in each growth period based on a multi-vegetation index in order to provide theoretical support for the application of remote sensing technology in cotton nutrition monitoring and diagnosis.

## 2. Materials and Methods

This study is divided into two parts, as shown in Figure 1. In the first part, according to the correlation between LAI, AGB, SLW, and LNC and CND in cotton, the hyperspectral vegetation indexes required for the study were selected. The correlation analysis between nitrogen and hyperspectral vegetation indexes for two years was used to determine the vegetation indexes for nitrogen estimation. In the second part of the study, MLR, PLSR, and SVR were used to establish cotton nitrogen estimation models in each growth period.

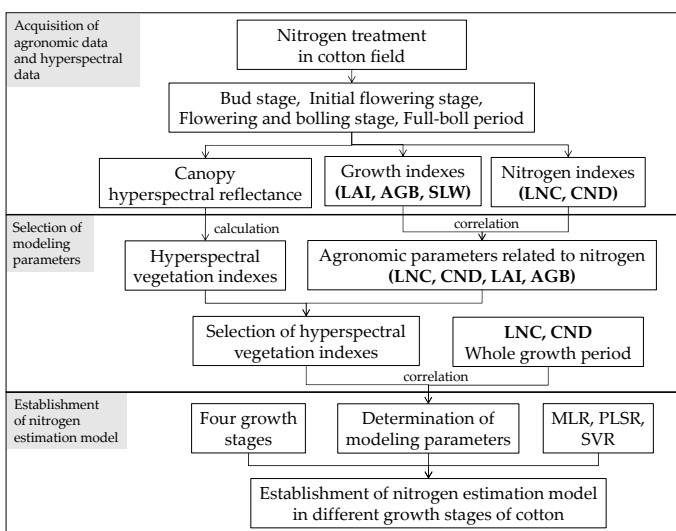

**Figure 1.** Flow chart illustrates important steps employed in this study.

### 2.1. Experimental Design

The experiment was conducted in the second (85°59′41″ E, 44°19′54″ N) of the teaching experimental fields of Shihezi University from April 2019 to September 2020. The sunshine times from April to September 2019 and 2020 in the test area were 9.17 h/day and 9.43 h/day; the active accumulated temperatures from April to September 2019 and 2020 were 20.92 °C/day and 21.77 °C/day, and the frost-free periods were 171 and 168 d. The soil was a loam that contained organic matter at 19.06 g/kg, total nitrogen 12.8 mg/kg, available phosphorus (P) 20.8 mg/kg, and available potassium (K) 165.1 mg/kg at the depth of 0–20 cm in 2019.

In order to obtain cotton plants with different nitrogen contents in this experiment, six nitrogen application levels were set: N0 (0 kg/ha), N1 (120 kg / ha), N2 (240 kg/ha),

N3 (360 kg/ha), N4 (480 kg/ha), and NC (278 kg/ha). The application timing of nitrogen fertilizer (urea, nitrogen content 46%) was synchronized with the drip application timing of local farmers. Nitrogen fertilizers were delivered using urea to cotton plants through drip irrigation over eight applications. The specific schedule of fertilizer applications is shown in Table 1. The base fertilizer of the cotton field was 150 kg/hm$^2$ of calcium superphosphate and 150 kg/hm$^2$ of potassium sulfate.

**Table 1.** Application amount of integrated drip irrigation of water and fertilizer in field experiment.

| Date (2019) | Fertilizer Percent | Water Volume (m$^3$/m$^2$) | Date (2020) | Fertilizer Percent | Water Volume (m$^3$/m$^2$) |
|---|---|---|---|---|---|
| 4–29 | 0% | 0.022 | 4–30 | 0% | 0.022 |
| 5–02 | 0% | 0.030 | 5–05 | 0% | 0.030 |
| 6–14 | 5% | 0.033 | 6–15 | 5% | 0.033 |
| 6–22 | 10% | 0.060 | 6–24 | 10% | 0.061 |
| 6–30 | 15% | 0.051 | 7–05 | 15% | 0.051 |
| 7–09 | 20% | 0.045 | 7–14 | 20% | 0.043 |
| 7.18 | 25% | 0.049 | 7.20 | 25% | 0.049 |
| 7.25 | 12% | 0.045 | 7.27 | 12% | 0.045 |
| 8.03 | 8% | 0.042 | 8.02 | 8% | 0.042 |
| 8.12 | 5% | 0.034 | 8.12 | 5% | 0.034 |
| 8.18 | 0% | 0.037 | 8.19 | 0% | 0.037 |

In 2019, the tested varieties were Xinluzao 45 (type II fruit branch, light green of leaf color) and Xinluzao 53 (type I fruit branch, dark green of leaf color), and the tested variety was Xinluzao 53 in 2020. The central planting area was mulched with drip irrigation for six planting rows of cotton plants spaced 0.66 m apart and 0.1 m within the row (Figure 2). The planting density was $21.50 \times 10^4$ plants/ha. The plot area was 25 m$^2$. Each nitrogen application treatment was repeated three times for a total of 18 plots in a randomized block design. In 2019, the sowing date of cotton was April 24; the emergence date was May 2, and the topping date was July 9. In 2020, the sowing date was April 18; the emergence date was April 30, and the top pruning date was July 10. Prevention and control of diseases and pests, application of herbicides, etc., were managed according to the local field. The growth rates of cotton were similar in the two years, and each growth period was approximately synchronous.

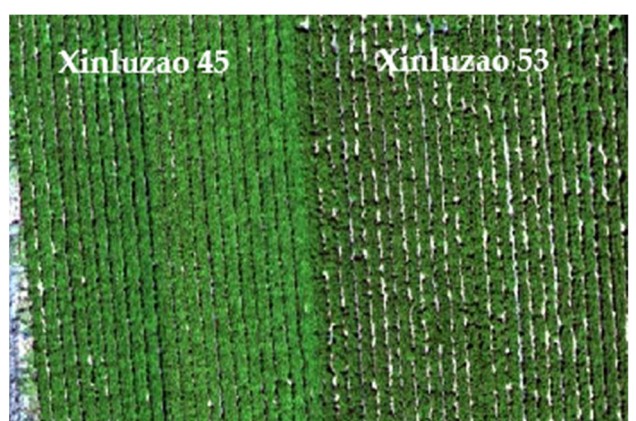
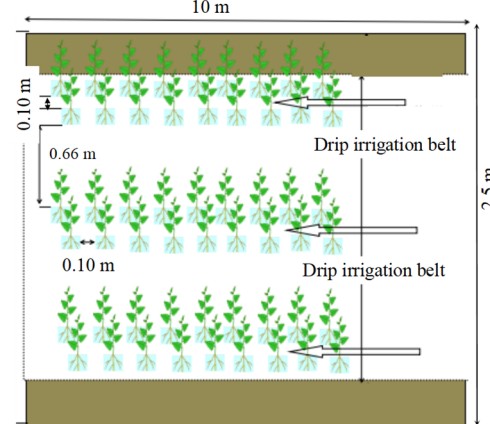

**Figure 2.** The top view of cotton test area (**left**) and internal row spacing configuration diagram of cotton field experimental community (**right**).

## 2.2. Data Acquisition

### 2.2.1. Sample Collection Time and Sample Size

In this study, the samples of leaf nitrogen content, aboveground biomass (AGB), leaf area index (LAI), and canopy hyperspectral reflectance in cotton were obtained according to the main reproductive period of cotton. The total sample size of the effect was 180; the total sample size of Xinluzao 45 (2019-45) was 180, and the total sample size of Xinluzao 53 (2019-53) was 156 in 2019. In 2020, the total sample size of Xinluzao53 (2020-53) was 180 (Table 2).

**Table 2.** Sample acquisition time and sample size.

| Year-Varieties | Item | Bud Stage | Initial Flowering Stage | Flowering Stage | Flowering and Bolling Stage | Full-Boll Period | Boll Opening Sage |
|---|---|---|---|---|---|---|---|
| 2019-45 | Date | 6–19 | 6–27 | 7–12 | 7–30 | 8–8 | // |
| | Sample size | 36 | 36 | 36 | 36 | 36 | |
| 2019-53 | Date | 6–20 | 7–6 | 7–12 | 7–30 | 8–8 | // |
| | Sample size | 36 | 30 | 36 | 18 | 36 | |
| 2020-53 | Date | 6–17 | 6–28 | // | 7–27 | 8–11 | 8–30 |
| | Sample size | 36 | 36 | | 36 | 36 | 36 |

Notes: // indicates that no samples were collected during this growth period.

### 2.2.2. Determination of Aboveground Biomass and Leaf Area Index in Cotton

Six cotton plants were selected for each experimental treatment and divided according to different organs of stem, leaf, bud, and boll. The plant leaf area was measured using an American LI-3100C area meter, and then the leaf area index (LAI) was calculated according to the plant leaf area [10]. After measuring the leaf area, all organs were heated for 30 min at 105 °C, then dried to a constant weight at 80 °C. The dry matter weight of each organ was calculated, and the sum of the biomasses of each organ represented the aboveground biomass (AGB) of the cotton plant [2].

### 2.2.3. Leaf Nitrogen Content and Canopy Nitrogen Density of Cotton

The nitrogen concentration in cotton leaves was determined by the Kjeldahl method. The dry leaf samples of the plant were crushed, sifted through a 100-mesh sieve, and the cotton leaf samples were digested with $H_2O_2–H_2SO_4$, using a 50 mL volume of the digestion solution. Then, a 10 mL aliquot was placed in a Kai nitrogen determiner for distillation (Haineng-K9840 automatic Kai nitrogen determiner), and the distilled solution was titrated with $(1/2)\ H_2SO_4$ to determine the nitrogen concentration (LNC) of cotton leaves.

Canopy nitrogen density (CND) is defined as the total leaf nitrogen per unit land area, $g/m^2$, which is calculated as follows [10]:

$$CND = LNC \times SLW \times LAI \tag{1}$$

where SLW is specific leaf weight, referring to the dry mass of leaves per unit leaf area, $g/cm^2$.

### 2.2.4. Acquisition of Hyperspectral Data of the Cotton Canopy

The hyperspectral data of the cotton canopy were obtained using an SR-3500 portable full-spectrum ground object spectrometer (Spectral Evolution Company, Lawrence, MA, USA) [21,22], and the parameters are shown in Table 3; we used the difference method to make the bandwidth of the instrument spectrum consistent, which was 1 nm.

**Table 3.** Technical parameters of SR-3500 portable full-spectrum ground object spectrometer.

| Technical Indicators | Parameter | Technical Indicators | Parameter |
|---|---|---|---|
| Spectral range | 350–2500 nm | Field of View | 25°; Integrating sphere |
| Spectral resolution | 3.5 nm (350–1000 nm) 10 nm (1000–1900 nm) 7 nm (1900–2500 nm) | Spectral bandwidth | 1.5 nm (350–1000 nm) 3.8 nm (1000–1900 nm) 2.5 nm (1900–2500 nm) |

The canopy spectra of drip-irrigated cotton were measured on sunny, cloudless, windless days or on days with low wind speed (≤grade 2), and the time range of the measurement day was controlled from 12:00 to 16:00. During measurement, the sensor probe was pointed vertically downward, and the vertical height from the cotton canopy top was about 1 m (Figure 3). Six points were collected for each experimental treatment, and five spectral data values were collected for each point. Finally, the average value of the five data points was used as the canopy spectral value of the sampling point of the plot. To ensure the measurement accuracy and reduce the influence of cloud and solar height changes on spectral reflectance, standard whiteboard correction was carried out for each group of targets before and after observation.

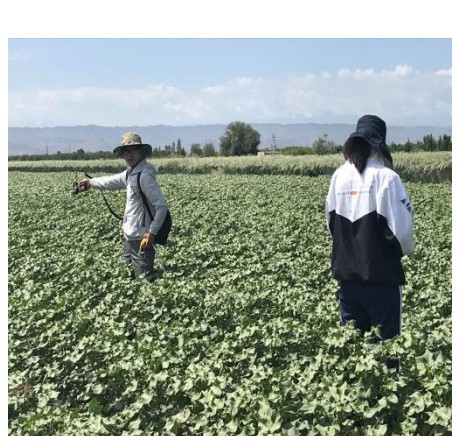
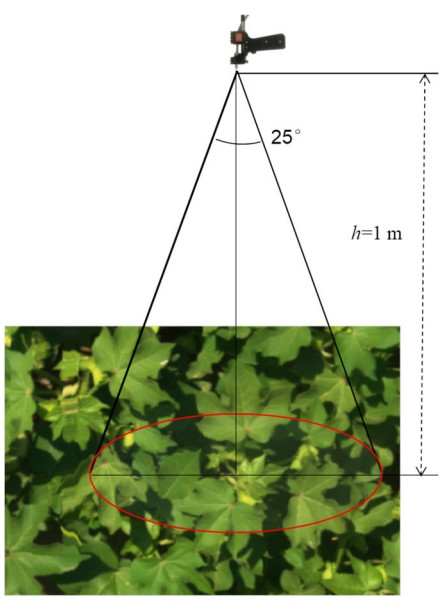
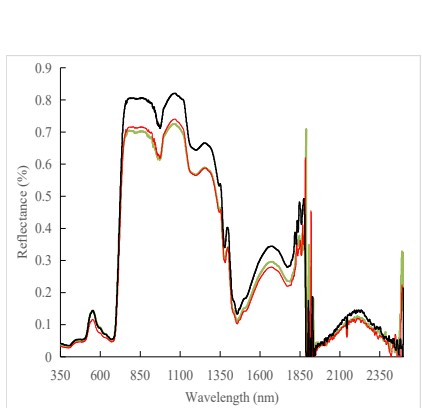

**Figure 3.** Acquisition of hyperspectral reflectance data of cotton canopy (2019-45, 19 June).

*2.3. Hyperspectral Vegetation Indexes Selected in This Research*

When crops lack nitrogen, a series of changes will occur in the nitrogen, chlorophyll, leaf area, and biomass of the plant. Because there are a large number of spectral indexes that can be used to estimate nitrogen content, leaf area, and biomass, this study referred to the relevant literature and selected 30 indexes with clear physical significance and high recognition for comparative analysis. The specific calculation methods and literature sources of each index are shown in Table 4.

**Table 4.** List of hyperspectral vegetation indexes used in this study.

| | Generic Name (Abbreviation) | Formula | Literature Source |
|---|---|---|---|
| Chlorophyll/nitrogen | Normalized Difference Vegetation Index (NDVI) | $(R800 - R680)/(R800 + R680)$ | [23] |
| | Difference Vegetation Index (DVI) | $R800 - R680$ | [23] |
| | Structure Insensitive Pigment Index (SIPI) | $(R800 - R445)/(R800 + R680)$ | [24] |
| | Modified red-edge normalized difference vegetation index (mND705) | $(R750 - R705)/(R750 + R705 - 2R445)$ | [25] |
| | Simple Ratio Index (SRI-1) | $R800/R680$ | [25] |
| | Modified Simple Ratio Index (MSRI) | $(R750 - R445)/(R705 - R445)$ | [25] |
| | The MERIS terrestrial Chlorophyll Index (MTCI) | $(R750 - R710)/(R710 - R680)$ | [23] |
| | Modified Chlorophyll Absorption in Reflectance Index (MCARI) | $[(R700 - R670) - 0.2 \times (R700 - R550)] \times (R700/R670)$ | [26] |
| | Optimized Soil-adjusted Vegetation Index (OSAVI) | $(R800 - R670)/(R800 + R670 + 0.16)$ | [27] |
| | Transformed Chlorophyll Absorption in Reflectance Index (TCARI) | $3 \times [(R700 - R670) - 0.2 \times (R700 - R550)] \times (R700/R670)$ | [28] |
| | Enhanced Vegetation Index (EVI) | $2.5 \times (R800 - R680)/(R800 + 6 \times R680 - 7.5 \times R450 + 1)$ | [29] |
| | Atmospherically Resistant Vegetation Index (ARVI) | $(R800 - 2 \times R680 + R450)/(R800 + 2 \times R680 - R450)$ | [29] |
| | Vogelmann Red-Edge Index 1 (VOG1) | $R740/R720$ | [30] |
| | Vogelmann Red-Edge Index 2 (VOG2) | $(R734 - R747)/(R715 + R726)$ | [31] |
| | Vogelmann Red-Edge Index 3 (VOG3) | $(R734 - R747)/(R715 + R720)$ | [31] |
| | Photochemical Reflectance Index (PRI) | $(R531 - R570)/(R531 + R570)$ | [32] |
| | PRI and red-edge Chlorophyll Index (PRI*CI) | $(R531 - R570)/(R531 + R570) \times (R760/R700 - 1)$ | [33] |
| | Plant senescence reflectance index (PSRI) | $(R678 - R500)/R750$ | [34] |
| | Carotenoid/chlorophyll ratio index (CCRI) | $[(R720 - R521) \times R705]/[(R750 - R705) \times R521]$ | [23] |
| | Simple ratio vegetation index (SRI-2) | $R515/R570$ | [35] |
| | Chlorophyll index in red-edge (CIred-edge) | $R800/R720 - 1$ | [36] |
| AGB | Normalized Dry Matter Index (NDMI) | $(R1650 - R1722)/(R1650 + R1722)$ | [7] |
| | Normalized Difference Tillage Index (NDTI) | $(R1650 - R2215)/(R1650 + R2215)$ | [37] |
| | 705nm Normalized Difference Vegetation (NDVI705) | $(R750 - R705)/(R750 + R705)$ | [38] |
| | Cellulose Absorption Index (CAI) | $100 \times [0.5 \times (R2010 + R2211) - R2101]$ | [26] |
| LAI | Modified triangular vegetation index 1 (MTVI 1) | $[120 \times (R800 - R550) - 2.5 \times (R670 - R550)]$ | [39] |
| | Modified triangular vegetation index 2 (MTVI 2) | $\dfrac{1.5 \times [1.2 \times (R800 - R550) - 2.5 \times (R670 - R550)]}{\sqrt{(2 \times R800 + 1)^2 - (6 \times R800 - 5 \times \sqrt{R670}) - 0.5}}$ | [39] |
| | Transformed triangular vegetation index (TTVI) | $0.5 \times [(783 - 740) \times (R865 - R740) - (865 - 740) \times (R783 - R740)]$ | [39] |
| | Standardized LAI-determining index (sLAIDI*) | $s \times ((R1050 - R1250)/(R1050 + R1250) \times R1555, s = 1$ | [19] |
| | The linear interpolation of red-edge inflection point (REIPlinear) | $700 + 40 \times [(Rred\text{-}edge - R700)/(R740 - R700)]$ $Rred\text{-}edge = (R670 - R780)/2$ | [40] |

### 2.4. Model Establishment Method and Model Evaluation Index

The concentration gradient method (2:1) was used to divide the data of each growth period and the entire growth period for 2019-45, 2019-53, and 2020-45 into a calibration set and a verification set. Based on the selection of the vegetation index, multiple linear regression (MLR), partial least-squares regression (PLSR), and support vector regression (SVR) were used to construct the model. MLRA and PLSR were implemented in the software the Unscrambler X 10.4 (2016, CAMO Analytics company, Trondheim, Norway), and SVR was implemented in the software MATLAB R2021b.

The performance of the models was evaluated by several indicators, including the test set determination coefficient ($R^2c$) and verification set determination coefficient ($R^2v$), the test set root mean square error (RMSEc) and verification set root mean square error (RMSEv), and the predicted relative standard deviation (RPD). The best model should have $R^2c$ and $R^2v$ values close to 1, low values of RMSEc and RMSEv, and a high RPD value.

### 3. Result

### 3.1. Statistical Analysis of the LNC in Cotton

### 3.1.1. Data Distribution Characteristics of Nitrogen and Biomass, Leaf Area Index, and Specific Leaf Weight of Drip-Irrigation Cotton

Table 5 shows the sample characteristics of various indicators of drip-irrigation cotton from 2019 to 2020. Among the two-year data, LAI was the most stable, with little difference in mean or standard deviation (SD) and with the ranges 2.813~2.948 and 1.336~1.691, respectively. For biomass, Xinluzao 53 was greater than that of Xinluzao 45, and the values of LNC and CND were higher than those of Xinluzao 45, indicating that the nitrogen demand of Xinluzao 53 was higher than that of Xinluzao 45 in the processes of growth and development. LNC and CND for Xinluzao 53 were lower in 2020, despite a significant

increase in the biomass. Indeed, max value (49.846 g/kg) of LNC is higher in 2020, but mean value (31.963 g/kg) of LNC and mean SD (2.713) of CND remain lower in 2020. The reason for this phenomenon was that the sufficient temperature and sunshine hours during cotton growth in 2020 were higher than those in 2019, resulting in the larger leaf area and biomass of cotton compared with 2019. However, the leaf nitrogen content per unit did not increase significantly (mainly controlled by cotton genotypes), which led to the low value of CND. From the analysis of interannual data, there were certain differences in canopy nitrogen density between years and varieties. From the existing data, the leaf nitrogen concentrations LNC, CND, and AGB were different between years. For Xinluzao 53, the average value of LNC in 2019 was 19.46% higher than in 2020, but the maximum value of LNC (49.846) and the SD value (7.306) in 2020 were greater than those in 2019, indicating that the LNC data in 2020 were more discrete than in 2019. The data characteristics of specific leaf weight were similar to those of leaf nitrogen concentration. The data dispersion in 2020 was greater than that in 2019, and there were some differences among varieties.

**Table 5.** Overall characteristics of samples in the entire growth period.

| Index (unit) | Varieties | Sample Number | Mean | Maximum | Minimum | SD | RSD |
|---|---|---|---|---|---|---|---|
| LNC (g/kg) | 2019-45 | 180 | 38.182 | 45.815 | 26.547 | 3.904 | 0.102 |
| | 2019-53 | 156 | 38.292 | 46.928 | 29.262 | 3.595 | 0.094 |
| | 2020-53 | 180 | 31.963 | 49.846 | 16.318 | 7.306 | 0.229 |
| CND (g/m$^2$) | 2019-45 | 180 | 8.247 | 21.737 | 1.995 | 4.469 | 0.542 |
| | 2019-53 | 156 | 9.418 | 22.099 | 2.509 | 3.71 | 0.394 |
| | 2020-53 | 180 | 7.242 | 18.991 | 3.034 | 2.713 | 0.375 |
| LAI | 2019-45 | 180 | 2.813 | 7.597 | 0.666 | 1.691 | 0.601 |
| | 2019-53 | 156 | 2.919 | 6.849 | 0.741 | 1.308 | 0.448 |
| | 2020-53 | 180 | 2.948 | 7.431 | 0.725 | 1.336 | 0.453 |
| AGB (t/ha) | 2019-45 | 180 | 5.833 | 19.09 | 0.87 | 4.651 | 0.797 |
| | 2019-53 | 156 | 7.003 | 27.314 | 1.219 | 4.498 | 0.642 |
| | 2020-53 | 180 | 9.692 | 32.340 | 1.319 | 6.509 | 0.672 |
| SLW (g/m$^2$) | 2019-45 | 180 | 79.989 | 105.687 | 53.051 | 7.582 | 0.095 |
| | 2019-53 | 156 | 87.871 | 113.07 | 58.401 | 8.030 | 0.091 |
| | 2020-53 | 180 | 86.300 | 196.303 | 42.84 | 26.016 | 0.301 |

### 3.1.2. Correlation between Nitrogen and AGB, LAI, and SLW at Different Growth Stages of Cotton

Nitrogen is the main factor affecting canopy indexes of cotton. This study considered cotton leaf nitrogen concentration (LNC) and canopy nitrogen density (CND) as nitrogen in cotton indexes; we studied the correlations with canopy indexes (AGB, LAI, and SLW) and analyzed the nutritional status of two cotton varieties in each growth period in 2019. It can be seen from Figure 4 that the correlation between LNC and SLW, AGB, and LAI of the two varieties of drip-irrigation cotton did not reach a significant level at any growth stage.

CND and AGB, and LAI of cotton showed extremely significant correlations in each growth period. In 2019, the correlations between CND and LAI of the two varieties reached the maximum values at the beginning of flowering, 0.935 and 0.944, respectively, and the correlations between CND and AGB reached the maximum values at the bud stage, 0.941 and 0.925, respectively (Figure 4). The correlations between CND and LAI and AGB of cotton at each growth stage were significantly higher than that of LNC, and the correlations were relatively stable. This was mainly because the nitrogen content of cotton leaves changed little during the entire growth period (Table 4), but AGB and LAI changed significantly with the growth process. This can be seen from Table 4 (RSD = 0.439~0.797).

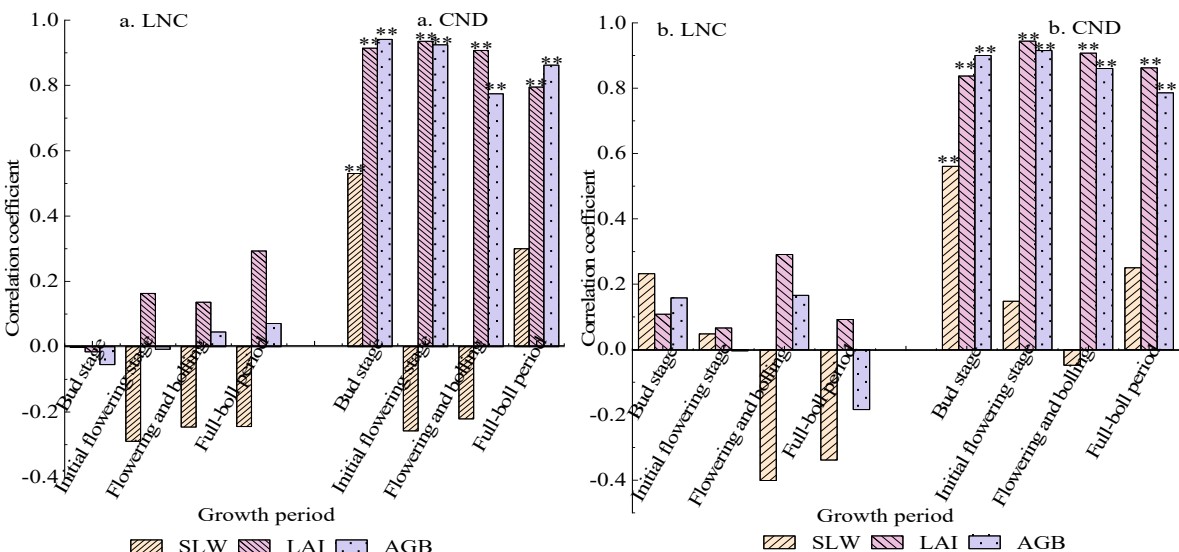

**Figure 4.** Correlations between nitrogen and AGB, LAI, and SLW at different growth stages of cotton in 2019 ((**a**): Xinluzao 45, (**b**): Xinluzao 53) (** indicates that significant difference is achieved under $p < 0.01$).

The correlation between SLW obtained from drip-irrigation cotton and nitrogen index in cotton showed that SLW and LNC showed a very weak correlation, and the significant correlation with CND was only in the cotton bud stage, with a maximum correlation coefficient of 0.565. There were significant differences among varieties. Therefore, the SLW index was not considered in the later spectral index analysis.

### 3.2. Correlation between Nitrogen in Cotton and Hyperspectral Vegetation Indexes

3.2.1. Correlation Analysis between Nitrogen in Cotton Indexes and Hyperspectral Vegetation Indexes at Different Growth Stages

By analyzing the correlations between nitrogen indexes LNC, CND, and vegetation indexes in each growth period of cotton (Figures 5 and 6), the highest correlation between LNC and vegetation index of Xinluzao 45 in 2019 appeared in the flowering and boll periods (Figure 5c). The vegetation index with the highest correlation coefficient was sLAIDI* (r = −0.617), followed by MTVI 1 (r = −0.574). The period with the highest correlation between canopy nitrogen density and vegetation index was the peak boll period. The vegetation index was NDTI, with Pearson's correlation coefficient r = 0.516, followed by sLAIDI*, with r = −0.501. The correlation between vegetation index and LNC and CND in other growth periods was weak, with r < 0.4.

In 2019, the highest correlation between LNC and vegetation index of Xinluzao 53 occurred in the peak boll period (Figure 6), and the index with the highest correlation coefficient was sLAIDI*, r = 0.400, followed by the NDMI index, with a correlation coefficient with LNC of −0.366. The vegetation index with the highest correlation with canopy nitrogen density was CCRI, r = 0.577, followed by REIP, r = −0.562, each of which appeared in the full-boll period (Figure 3d). In this growth period, the correlations of 14 hyperspectral vegetation indexes reached above 0.5.

3.2.2. Correlation Analysis between Vegetation Index and Nitrogen Index during the Entire Growth Period of Cotton

This study analyzed the correlation between nitrogen data and vegetation index in the entire growth period of Xinluzao 45 (2019-45, n = 180) and Xinluzao 53 (2019-53, n = 156) in 2019, and Xinluzao 53 (2020-53, n = 180) in 2020 (Table 6). The results showed that there were significant differences in the correlations between each vegetation index and cotton LNC and CND between years and varieties. By analyzing the correlations between LNC and vegetation index during the entire growth period, there were significant differences

between NDVI, MCARI, ARVI, PRI*CI, TTVI, sLAIDI*, NDMI, and NDTI for Xinluzao 53 between years (2019 and 2020). In particular, MCARI, TTVI, and NDMI showed extremely significant correlations with LNC in 2020 (r > 0.5), but they did not reach significance in 2019. The correlation coefficient r was only about 0.1, which may indicate that the correlation between this part of the vegetation index and LNC was vulnerable to the meteorological environment. The correlations between hyperspectral vegetation indexes MSRI, MTCI, MCARI, NDVI705, VOG1, VOG2, VOG3, CIred-edge, and MTVI 2 and LNC differed among varieties, and the correlations with 2019-45 were extremely significant, but the correlations with 2019-53 were very low or even zero, and the correlations between NDVI705, VOG1, VOG2, VOG3, and CIred-edge and LNC did not reach 0.1. In addition, the correlations between NDVI, OSAVI, ARVI, MTVI 1, TTVI, sLAIDI*, and CAI and LNC reached the significant level in two varieties in 2019, but the correlation was unstable.

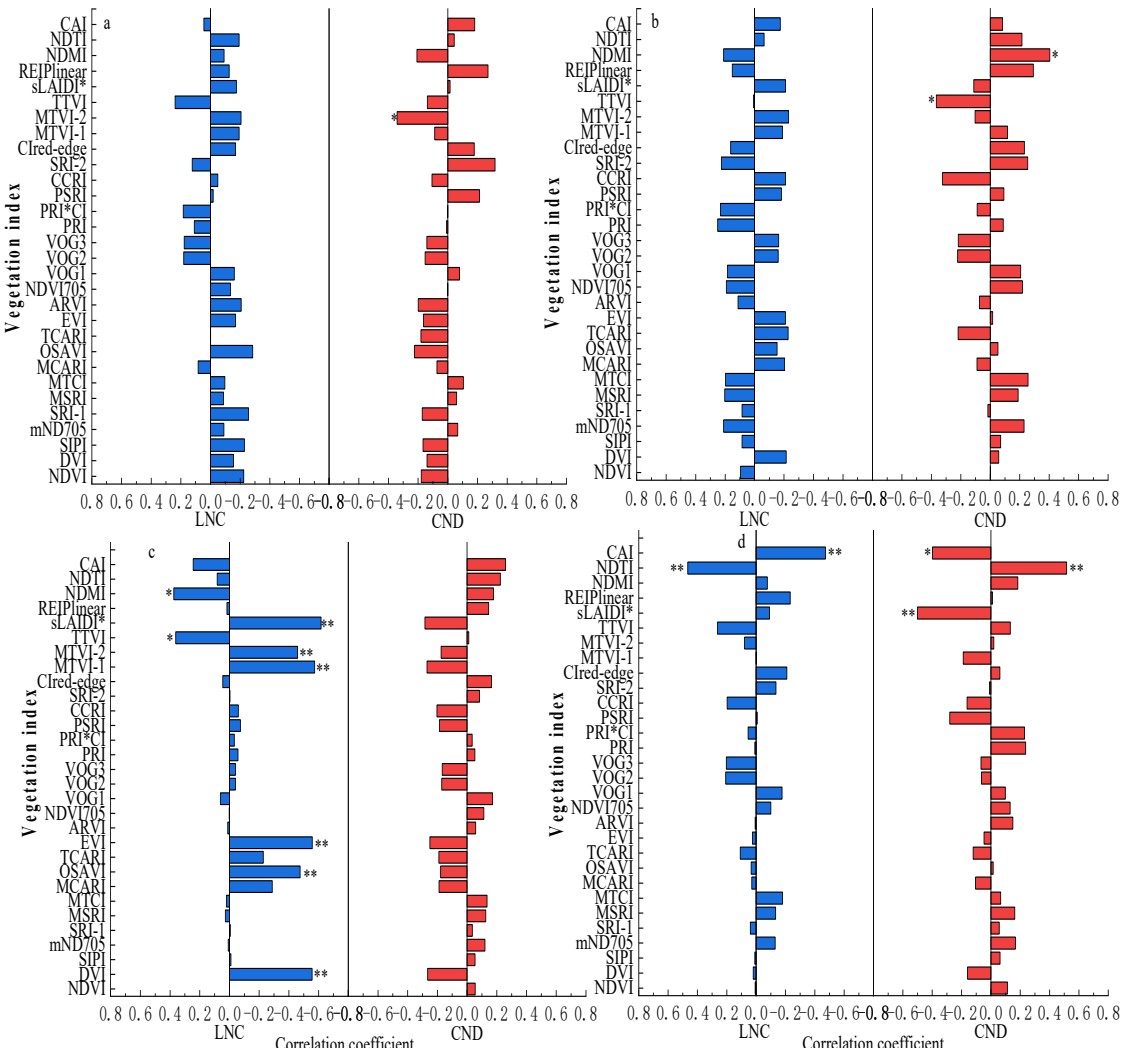

**Figure 5.** Correlations between LNC, CND, and vegetation index in 2019 (Xinluzao 45, (**a**): bud stage; (**b**): initial flowering stage; (**c**): flowering and boll stage; (**d**): full-boll period. * Indicates that significant difference is achieved under *p* < 0.05; ** indicates that significant difference is achieved under *p* < 0.01).

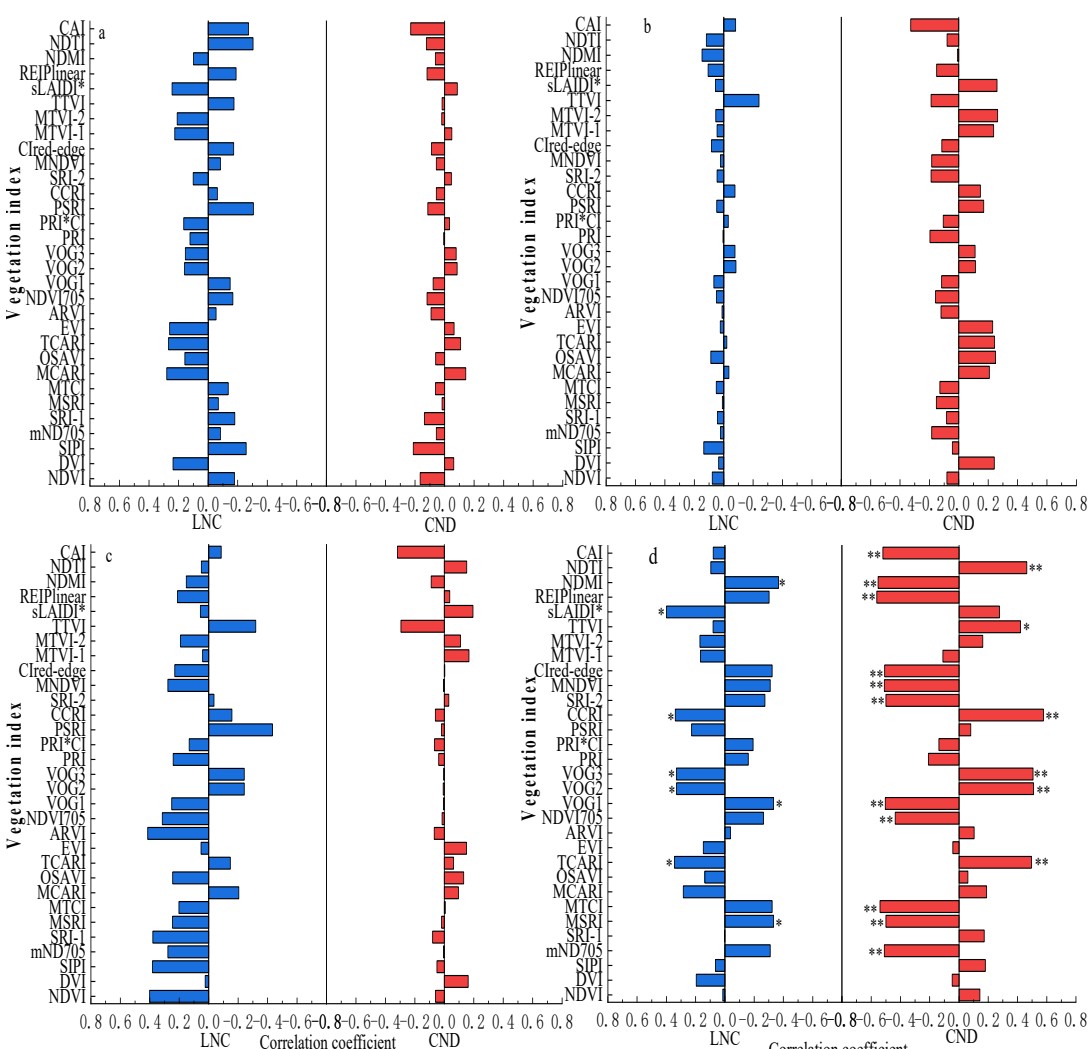

**Figure 6.** Correlation between LNC, CND, and vegetation index in 2019 (Xinluzao 53, (**a**): bud stage; (**b**): initial flowering stage; (**c**): flowering and boll stage; (**d**): full-boll period. * Indicates that significant difference is achieved under *p* < 0.05; ** indicates that significant difference is achieved under *p* < 0.01).

The correlation between CND and vegetation index was stronger. The correlations between MCARI, TCARI, and CND were different between years, and the correlation coefficients of the two varieties were greater than 0.40. The indexes mND705, MSRI, CCRI, and SRI-2 differed among varieties. Although the correlations reached a significant level in 2019, the correlation coefficient with Xinluzao 45 was more than 0.3 higher than that with Xinluzao 53. The correlations between indexes PRI and PRI*CI and CND differed between varieties and years. The correlations between indexes OSAVI, EVI, MTVI 1, MTVI 2, sLAIDI*, NDMI, CAI, and CND were unstable, and the correlations between the two varieties between years showed the opposite pattern.

There was an opposite correlation between the two varieties, indicating that there were significant differences between LNC and these hyperspectral vegetation indexes. Among the 30 hyperspectral vegetation indexes, TCARI, PRI, CCRI, SRI-2, and LNC had very significant correlations between years and varieties, and the correlations had a certain stability and consistency. Therefore, these four hyperspectral vegetation indexes were selected as the research objects with which to establish the estimation model of LNC for the entire growth period and each individual growth period.

**Table 6.** Correlation analysis between LNC, CND, and spectral index of cotton in different years.

| | **LNC** | | | | **CND** | | |
| --- | --- | --- | --- | --- | --- | --- | --- |
| | **2019-45** | **2019-53** | **2020-53** | | **2019-45** | **2019-53** | **2020-53** |
| NDVI | −0.224 ** | 0.219 ** | 0.019 | NDVI | 0.300 ** | 0.252 ** | 0.190 * |
| DVI | −0.106 | 0.283 ** | 0.547 ** | DVI | 0.151 * | −0.385 ** | 0.066 |
| SIPI | −0.240 ** | 0.197 * | −0.097 | SIPI | 0.317 ** | 0.332 ** | 0.177 * |
| mND705 | −0.388 ** | −0.090 | 0.156 * | mND705 | 0.581 ** | 0.269 ** | 0.329 ** |
| SRI-1 | −0.134 | 0.248 ** | 0.049 | SRI-1 | 0.206 ** | 0.255 ** | 0.189 * |
| MSRI | −0.373 ** | −0.096 | 0.089 | MSRI | 0.561 ** | 0.280 ** | 0.345 ** |
| MTCI | −0.397 ** | −0.130 | 0.012 | MTCI | 0.568 ** | 0.358 ** | 0.337 ** |
| MCARI | 0.322 ** | 0.144 | 0.487 ** | MCARI | −0.453 ** | −0.458 ** | −0.138 |
| OSAVI | −0.194 ** | 0.421 ** | 0.463 ** | OSAVI | 0.269 ** | −0.201 * | 0.157 * |
| TCARI | 0.387 ** | 0.251 ** | 0.384 ** | TCARI | −0.603 ** | −0.434 ** | −0.160 * |
| EVI | −0.116 | 0.292 ** | 0.564 ** | EVI | 0.171 * | −0.408 ** | 0.090 |
| ARVI | −0.198 ** | 0.232 ** | 0.126 | ARVI | 0.272 ** | 0.136 | 0.196 ** |
| NDVI705 | −0.356 ** | −0.004 | 0.102 | NDVI705 | 0.529 ** | 0.318 ** | 0.334 ** |
| VOG1 | −0.352 ** | −0.026 | −0.017 | VOG1 | 0.522 ** | 0.349 ** | 0.349 ** |
| VOG2 | 0.383 ** | 0.071 | 0.055 | VOG2 | −0.544 ** | −0.450 ** | −0.355 ** |
| VOG3 | 0.375 ** | 0.066 | 0.051 | VOG3 | −0.536 ** | −0.436 ** | −0.355 ** |
| PRI | −0.422 ** | −0.290 ** | −0.239 ** | PRI | 0.590 ** | 0.125 | 0.315 ** |
| PRI*CI | −0.242 ** | −0.357 ** | 0.145 | PRI*CI | 0.258 ** | 0.043 | 0.266 ** |
| PSRI | 0.021 | 0.039 | −0.311 ** | PSRI | −0.046 | −0.01 | −0.177 * |
| CCRI | 0.387 ** | 0.245 ** | 0.328 ** | CCRI | −0.640 ** | −0.309 ** | −0.290 ** |
| SRI-2 | −0.533 ** | −0.432 ** | −0.357 ** | SRI-2 | 0.626 ** | 0.216 ** | 0.070 |
| CIred−edge | −0.390 ** | −0.099 | −0.121 | CIred−edge | 0.554 ** | 0.458 ** | 0.329 ** |
| MTVI 1 | −0.181 * | 0.279 ** | 0.504 ** | MTVI 1 | 0.257 ** | −0.358 ** | 0.079 |
| MTVI 2 | −0.066 | 0.444 ** | 0.551 ** | MTVI 2 | 0.092 | −0.285 ** | 0.125 |
| TTVI | 0.345 ** | −0.144 | −0.534 ** | TTVI | −0.497 ** | −0.231 ** | −0.268 ** |
| sLAIDI* | −0.371 ** | 0.159 * | 0.566 ** | sLAIDI* | 0.437 ** | −0.211 ** | 0.054 |
| REIPlinear | −0.443 ** | −0.192 * | −0.052 | REIPlinear | 0.619 ** | 0.530 ** | 0.320 ** |
| NDMI | −0.212 ** | −0.156 | −0.672 ** | NDMI | 0.381 ** | 0.007 | −0.025 |
| NDTI | 0.112 | −0.114 | −0.215 ** | NDTI | −0.067 | 0.408 ** | 0.143 |
| CAI | −0.235 ** | 0.202 * | −0.110 | CAI | 0.272 ** | −0.480 ** | 0.037 |

Notes: * indicates that significant difference is achieved under *p* < 0.05; ** indicates that significant difference is achieved under *p* < 0.01.

The correlations between mND705, SRI-1, MSRI, MTCI, TCARI, NDVI705, VOG1, VOG2, VOG3, CCRI, CIred-edge, REIPlinear, and CND reached significant levels between years and varieties, and the correlations were relatively stable and consistent. Therefore, the indexes with good stability (MTCI, NDVI705, VOG1, VOG2, VOG3, CIred-edge, and REIPlinear) were selected to establish the estimation model for the entire growth period and each individual growth period of CND.

3.2.3. Correlation Analysis between Hyperspectral Vegetation Index and AGB and LAI during the Entire Growth Period of Cotton

The correlations between LAI and AGB and vegetation index are shown in Table 7. The correlations between DVI, OSAVI, EVI, MTVI 1, MTVI 2, sLAIDI*, and LAI differed among varieties. The index with the largest difference was DVI. The correlation coefficient between DVI with LAI was −0.422 in Xinluzao 53 of 2019, but the correlation coefficient with Xinluzao 45 of 2019 was only 0.131. The correlations between NDVI, SIPI, mND705, SRI-1, MSRI, MTCI, MCARI, TCARI, NDVI705, VOG1, VOG2, VOG3, PRI, SRI-2, CIred-edge, and REIPlinear (16 indexes) and LAI reached significant levels (*p* < 0.05) between years and varieties, and the correlations were relatively stable.

**Table 7.** Correlations between spectral indexes among different years and varieties and LAI and AGB of cotton.

| | LAI | | | | AGB | | |
| --- | --- | --- | --- | --- | --- | --- | --- |
| | **2019-45** | **2019-53** | **2020-53** | | **2019-45** | **2019-53** | **2020-53** |
| NDVI | 0.282 ** | 0.163 * | 0.234 ** | NDVI | 0.152 * | 0.048 | 0.005 |
| DVI | 0.131 | −0.422 ** | −0.267 ** | DVI | −0.044 | −0.413 ** | −0.544 ** |
| SIPI | 0.299 ** | 0.249 ** | 0.299 ** | SIPI | 0.166 * | 0.126 | 0.092 |
| mND705 | 0.550 ** | 0.293 ** | 0.272 ** | mND705 | 0.428 ** | 0.214 ** | 0.003 |
| SRI-1 | 0.186 ** | 0.163 * | 0.211 ** | SRI-1 | 0.070 | 0.053 | −0.040 |
| MSRI | 0.530 ** | 0.297 ** | 0.329 ** | MSRI | 0.414 ** | 0.235 ** | 0.065 |
| MTCI | 0.552 ** | 0.389 ** | 0.374 ** | MTCI | 0.448 ** | 0.324 ** | 0.131 |
| MCARI | −0.439 ** | −0.463 ** | −0.476 ** | MCARI | −0.415 ** | −0.391 ** | −0.515 ** |
| OSAVI | 0.246 ** | −0.296 ** | −0.097 | OSAVI | 0.077 | −0.373 ** | −0.433 ** |
| TCARI | −0.578 ** | −0.481 ** | −0.414 ** | TCARI | −0.538 ** | −0.423 ** | −0.473 ** |
| EVI | 0.150 * | −0.449 ** | −0.251 ** | EVI | −0.024 | −0.440 ** | −0.541 ** |
| ARVI | 0.255 ** | 0.048 | 0.166 * | ARVI | 0.129 | −0.049 | −0.075 |
| NDVI705 | 0.499 ** | 0.309 ** | 0.320 ** | NDVI705 | 0.369 ** | 0.204 * | 0.035 |
| VOG1 | 0.499 ** | 0.344 ** | 0.403 ** | VOG1 | 0.382 ** | 0.254 ** | 0.149 * |
| VOG2 | −0.527 ** | −0.459 ** | −0.442 ** | VOG2 | −0.423 ** | −0.383 ** | −0.176 * |
| VOG3 | −0.518 ** | −0.443 ** | −0.438 ** | VOG3 | −0.414 ** | −0.367 ** | −0.173 * |
| PRI | 0.580 ** | 0.214 ** | 0.216 ** | PRI | 0.457 ** | 0.241 ** | −0.073 |
| PRI*CI | 0.280 ** | 0.148 | 0.210 ** | PRI*CI | 0.272 ** | 0.223 ** | 0.000 |
| PSRI | −0.047 | −0.034 | −0.023 | PSRI | −0.009 | −0.067 | 0.189 * |
| CCRI | −0.609 ** | −0.389 ** | −0.118 | CCRI | −0.508 ** | −0.365 ** | 0.143 |
| SRI-2 | 0.628 ** | 0.348 ** | 0.297 ** | SRI-2 | 0.686 ** | 0.410 ** | 0.445 ** |
| CIred−edge | 0.537 ** | 0.475 ** | 0.455 ** | CIred−edge | 0.438 ** | 0.404 ** | 0.236 ** |
| MTVI 1 | 0.234 ** | −0.393 ** | −0.225 ** | MTVI 1 | 0.060 | −0.392 ** | −0.500 ** |
| MTVI 2 | 0.073 | −0.386 ** | −0.192 ** | MTVI 2 | −0.099 | −0.442 ** | −0.522 ** |
| TTVI | −0.470 ** | −0.225 ** | 0.026 | TTVI | −0.314 ** | −0.148 | 0.460 ** |
| sLAIDI* | 0.430 ** | −0.210 ** | −0.283 ** | sLAIDI* | 0.310 ** | −0.154 | −0.583 ** |
| REIPlinear | 0.602 ** | 0.582 ** | 0.404 ** | REIPlinear | 0.506 ** | 0.530 ** | 0.182 * |
| NDMI | 0.363 ** | 0.015 | 0.362 ** | NDMI | 0.233 ** | 0.007 | 0.733 ** |
| NDTI | −0.070 | 0.396 ** | 0.316 ** | NDTI | −0.184 ** | 0.380 ** | 0.235 ** |
| CAI | 0.258 ** | −0.477 ** | −0.030 | CAI | 0.212 ** | −0.484 ** | 0.128 |

Notes: * indicates that significant difference is achieved under $p < 0.05$; ** indicates that significant difference is achieved under $p < 0.01$.

The correlation between AGB and vegetation index showed that the correlations with DVI and EVI index showed differences among varieties, but the correlation difference between years was inconsistent. Compared with the correlation between LAI and hyperspectral vegetation indexes of the band with red-edge, such as MSRI, MTCI, VOG1, VOG2, VOG3, CIred-edge, and REIPlinear, the correlation between AGB and vegetation index was weak. The hyperspectral vegetation indexes with good stability and strong correlations were TCARI and SRI-2. The correlation between these two indexes and the AGB of Xinluzao 45 was as high as 0.686 in 2019, showing a very significant correlation. Relative to Xinluzao 45, the correlation coefficients between Xinluzao 53 and the indexes were low, ranging from 0.40 to 0.45.

Through a comprehensive comparative analysis of the correlations between LNC, CND, LAI, AGB, and 30 hyperspectral vegetation indexes of drip-irrigation cotton, the correlation between LNC and vegetation index was the least, and the highest correlation index was SRI-2, r = −0.533. There was also a very significant correlation between this index and LAI and AGB between years and varieties, with the highest values being 0.628 and 0.686, respectively. The correlation of LNC as an individual index was lower than that of the other three population canopy indexes (CND, AGB, LAI). It showed that canopy hyperspectral is more suitable for quantitative analysis of population canopy indexes. Secondly, the canopy indexes CND, LAI, and AGB were correlated with the hyperspectral vegetation indexes CIred-edge and REIPlinear. These two indexes were related to the

red-edge position, indicating that the red-edge position has a high potential value for studying the canopy indexes of drip-irrigation cotton.

### 3.3. Establishment and Validation of the LNC and CND in Cotton Estimation Model Based on Hyperspectral Vegetation Indexes

The modeling methods of this study were simple multiple linear regression (MLR), partial least-squares regression (PLSR), and support vector machine regression (SVR). The hyperspectral vegetation indexes obtained from the analysis in 3.2.2 were used to establish the estimation model of each index of each growth period and for the entire growth period of drip-irrigation cotton. The model parameters of LNC were TCARI, PRI, CCRI, and SRI-2, and the model parameters of CND were MTCI, NDVI705, VOG1, VOG2, VOG3, CIred-edge, and REIPlinear. The best estimation models for the entire growth period and individual growth periods were obtained by using the coefficient of determination ($R^2$) and the RPD optimality principle.

#### 3.3.1. Establishment and Verification of the Nitrogen in Cotton Entire Growth Period Estimation Model Based on Hyperspectral Vegetation Index

The model parameters of the LNC and CND of drip-irrigation cotton are shown in Table 8. The $R^2$ of the estimation model of leaf nitrogen concentration in 2020-53 was higher, and the coefficient of determination of the model had the range $R^2 = 0.442 \sim 0.797$. For 2019-45, the model accuracy was less than for 2020-53, but its RMSE value was the smallest. For the two-year model, the verification model $R^2$ of 2019-45 was the lowest. Among the three modeling methods, the precision of the support vector machine regression method in 2020-53 was the highest; $R^2c = 0.797$, $R^2v = 0.612$. For the LNC of cotton in this study, SVR could improve the accuracy of the estimation model.

**Table 8.** Parameters of the nitrogen estimation model for the entire growth period of drip-irrigation cotton based on vegetation index.

| Nitrogen Index | Model Parameter | 2020-53 | | | 2019-53 | | | 2019-45 | | |
|---|---|---|---|---|---|---|---|---|---|---|
| | | MLR | PLSR | SVR | MLR | PLSR | SVR | MLR | PLSR | SVR |
| LNC | $R^2c$ | 0.442 | 0.442 | 0.797 | 0.245 | 0.239 | 0.259 | 0.36 | 0.346 | 0.705 |
| | RMSEc | 5.569 | 5.476 | 3.308 | 3.198 | 3.134 | 3.100 | 3.193 | 3.223 | 2.241 |
| | $R^2v$ | 0.769 | 0.520 | 0.612 | 0.680 | 0.362 | 0.289 | 0.179 | 0.172 | 0.239 |
| | RMSEv | 4.988 | 4.988 | 4.929 | 2.871 | 2.851 | 3.026 | 3.535 | 3.552 | 3.417 |
| CND | $R^2c$ | 0.207 | 0.113 | 0.197 | 0.590 | 0.565 | 0.708 | 0.600 | 0.574 | 0.770 |
| | RMSEc | 2.536 | 2.579 | 2.501 | 2.498 | 2.472 | 3.652 | 2.923 | 2.912 | 2.144 |
| | $R^2v$ | 0.586 | 0.090 | 0.118 | 0.742 | 0.403 | 0.406 | 0.555 | 0.562 | 0.642 |
| | RMSEv | 2.491 | 2.517 | 2.502 | 2.693 | 2.775 | 3.664 | 0.138 | 1.020 | 1.180 |

For the establishment of the estimation model of cotton canopy nitrogen density under drip irrigation, the estimation model of 2020-53 had the least effect, with the highest $R^2c$ of 0.113, but the model $R^2c$ of 2019-45 was the highest (0.770); $R^2v = 0.642$, and the RMSEc was higher than the model RMSEc established by SVR for 2019-53. The overall modeling effect of 2019-45 was better; $R^2 = 0.574 \sim 0.770$. Among the three modeling methods, SVR had the best effect. Combined with the minimum value of CND in Table 4 (overall characteristics of samples in the whole growth period) and the RMSE of the model, the results indicated that the established model was not conducive to estimating the smaller value of CND.

#### 3.3.2. Establishment and Verification of a Cotton Leaf Nitrogen Concentration Estimation Model at Different Growth Stages Based on Vegetation Index

In this study, MLR, PLSR, and SVR were used to estimate the leaf nitrogen concentration in four growth stages (bud stage, initial flowering stage, flowering and boll stage, and full-boll stage) of drip-irrigation cotton. As shown in Figure 7, SVR improved the accuracy of the model for leaf nitrogen concentration in different years, especially at the

initial flowering stage. The accuracy $R^2c$ of the estimation model was between 0.732~1. In the bud stage of 2019-45 and the initial flowering stage of 2019-53, the estimation model $R^2c$ values of the LNC were 0.990 and 1, but the validation model $R^2v$ values were 0.158 and 0.017, respectively. The overfitting phenomenon of the model showed that for the application of SVR, it was necessary to select an appropriate training set and validation set data division method to improve the applicability of the model. Compared with the overfitting of the SVR model, the model accuracy $R^2c$ obtained by the MLR modeling method was the highest, occurring in the flower and boll periods in 2019-53. The modeling accuracy was 0.609, and the verified $R^2v$ was 0.25. The overfitting phenomenon also occurred.

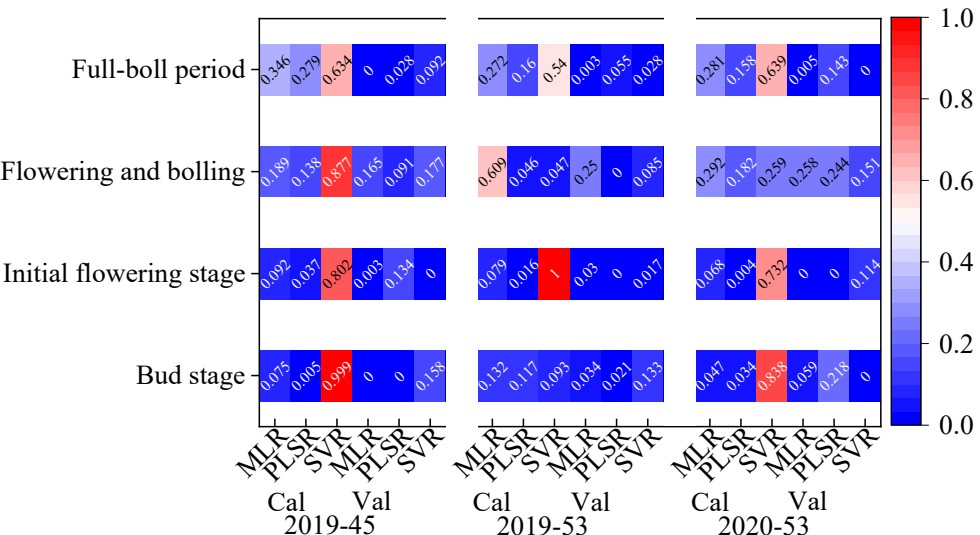

**Figure 7.** Estimation of model parameters of cotton leaf nitrogen concentration at different growth stages.

By analyzing the model parameters of cotton at each growth stage, the model accuracy at the bud stage was the lowest, and that at the flowering and boll stages was the highest (Figure 7). Compared with other growth stages, the model $R^2$ established by the test set and verification set in the flowering and boll stages was relatively stable, especially in the model parameters of 2020-53. The $R^2$ of the test set and verification set model was stable between 0.151 and 0.292, and the accuracy of the model was low. On the whole, the modeling effect of the vegetation index screened by correlation in each growth period of leaf nitrogen concentration of drip-irrigation cotton was lower than that for the entire growth period.

### 3.3.3. Establishment and Verification of a Cotton Canopy Nitrogen Density Estimation Model at Different Growth Stages Based on Hyperspectral Vegetation Indexes

According to the analysis in Table 7, seven hyperspectral vegetation indexes (MTCI, NDVI705, VOG1, VOG2, VOG3, CIred-edge, and REIPlinear) were selected to establish the CND model for cotton in each growth period. The MLR method used to estimate and verify the model of cotton bud stage under drip irrigation had the highest accuracy, especially the model established in 2020-53, with $R^2c$ of 0.427 and $R^2v$ of 0.406 (Figure 8). The model for the bud period had the highest accuracy via the MLR method, especially the model established in 2020-53 (Figure 8). The model $R^2c$ was 0.427, $R^2v$ was 0.406, and the established model was relatively stable. Compared with the parameter RPD of the model, the model R2 established by the PLSR method was about 0.19, but the RPDv was the highest among all validation models at 0.929. The $R^2c$ values of 2019-45, 2019-53, and 2020-53 bud-stage CND estimation models established by SVR were the highest (0.476, 1, and 0.877), but there was an overfitting phenomenon, and the prediction results could not be achieved for higher CND values.

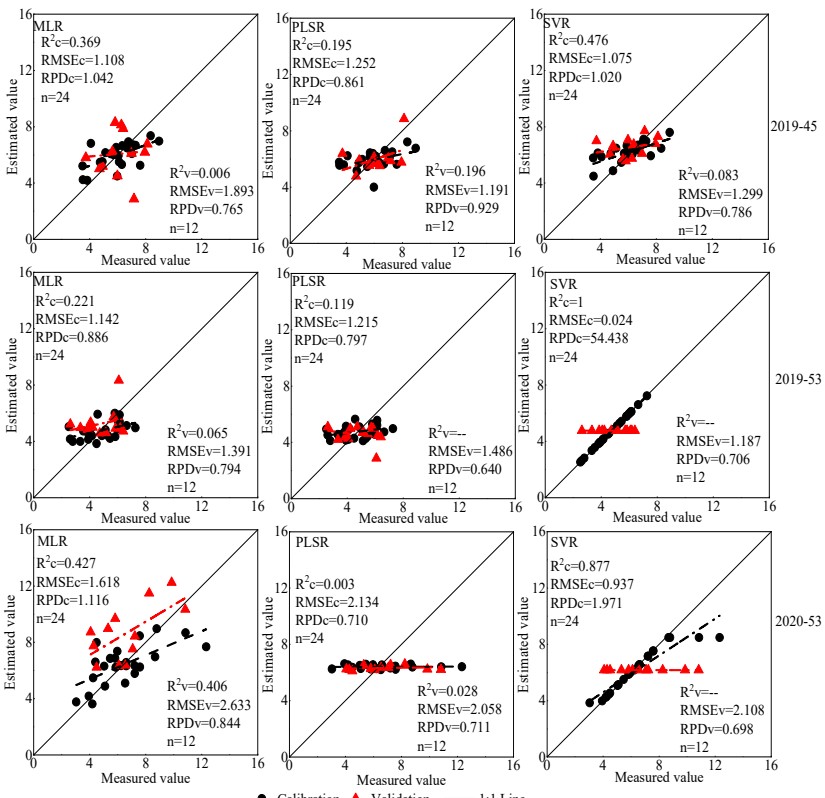

**Figure 8.** Establishment of the CND estimation model for cotton bud stage under drip irrigation.

Compared with the bud stage, the estimation accuracy of canopy nitrogen density in the initial flowering stage of drip-irrigation cotton was not improved (Figure 9), which was consistent with the results for the bud stage. The model established by SVR also had a serious overfitting phenomenon, and the canopy nitrogen density data in 2019-53 also had serious overfitting.

Compared with the bud stage and initial flowering stage, the accuracy of CND estimation models for the flowering and boll stages and full-boll stage was improved. The model parameters established by MLR in the flowering and boll stages showed that the $R^2c$ and $R^2v$ of the CND of cotton estimation models in 2019 and 2020 were stable at 0.544~0.658. Among these, 2020-53 had the best modeling effect; $R^2c$ and $R^2v$ were 0.658 and 0.601, respectively, and RPD values were 1.558 and 1.131 (Figure 10). However, the RMSEv value of the 2020-53 model validation set was higher than those of the two varieties in 2019, which were 2.127, 1.045, and 1.947, respectively. Among the two varieties in 2019 and 2020, the parameters RPDc and RPDv established by MLR were 1.131~1.840. The RPD value of the model established based on the data of 2019-53 was the highest, and the RMSE value was the lowest. The RPD of the test set and the verification set were 1.840 and 1.774, and the RMSE values were 1.123 and 1.045. In this study, the precision of the model established by the PLSR method for the flowering and boll stages was better than those for the bud stage and initial flowering stage. The $R^2c$ values of the model constructed by PLSR in 2019-45 and 2020-53 were 0.568 and 0.658, respectively. In general, the three modeling methods were applicable to the establishment of a cotton CND model in 2019-45; $R^2c$ = 0.568~0.764, RPDv = 1.155~1.284. The MLR modeling method was suitable for the estimation of cotton CND between different years and varieties.

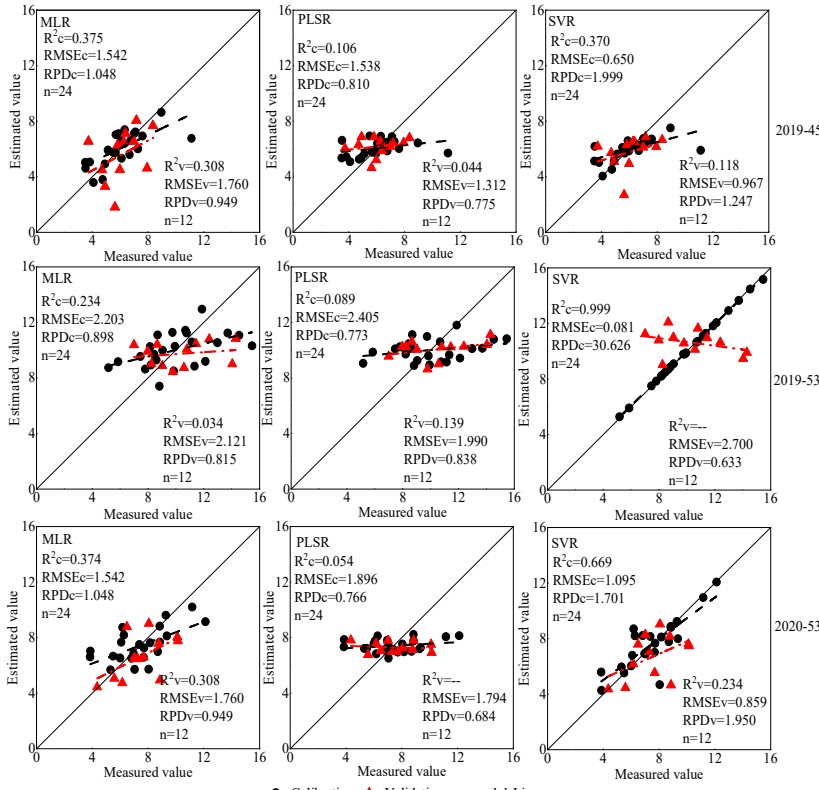

**Figure 9.** Vegetation index estimation and validation model parameters of canopy nitrogen density of drip-irrigation cotton at the initial flowering stage.

According to the establishment of the CND estimation model in the full-boll period of drip-irrigation cotton (Figure 11), the $R^2$ value of the estimation and verification model was the highest and the most stable. The $R^2c$ and $R^2v$ values were 0.558 and 0.600, and RPDc and RPDv were 1.348 and 1.284, respectively. In the full-boll period, the modeling effects of PLSR and SVR did not show advantages, and overfitting occurred in the CND estimation by SVR in four growth stages.

To summarize, when using multiple hyperspectral vegetation indexes to estimate LNC and CND of cotton in individual growth periods, the MLR method was the most stable, the estimation established by SVR was prone to overfitting, and the estimation accuracy of the CND model was better than that of the LNC model. The results of this study showed that in the four different growth periods, the estimation effect of CND in the later growth period (flowering and boll period and full-boll period) was the best. The vegetation index calculated by the original canopy spectrum was used in this study. The planting density of drip-irrigation cotton in Northern Xinjiang was relatively high. In the later growth period of cotton, the ridge of cotton is closed, reducing the impact of soil and plastic film on the canopy spectrum. The hyperspectral vegetation indexes could better reflect the nutritional status of cotton.

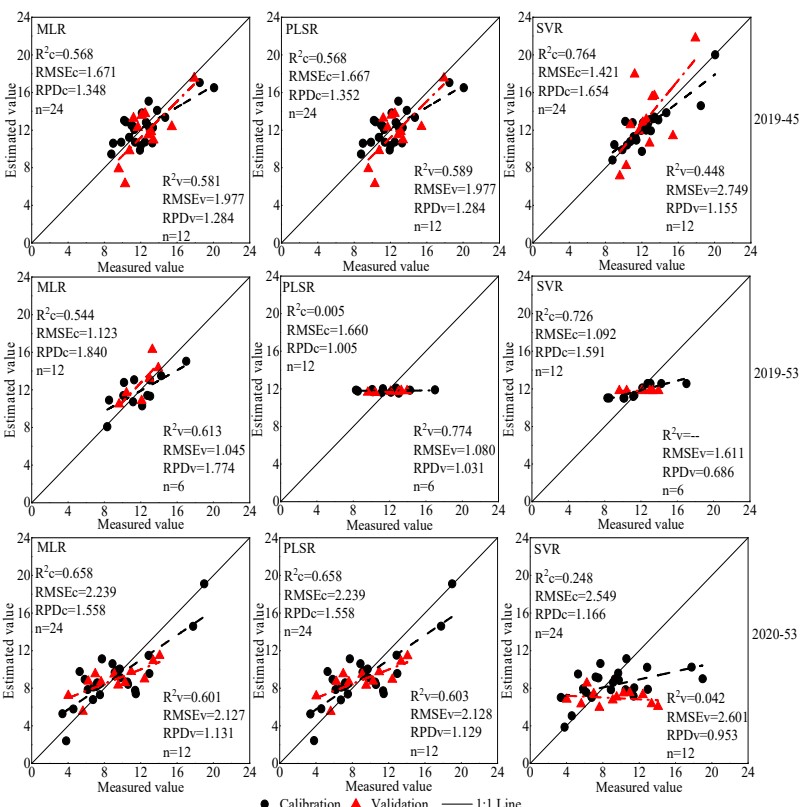

**Figure 10.** Parameters of CND estimation and verification model of drip-irrigation cotton at flowering and boll stage based on vegetation index.

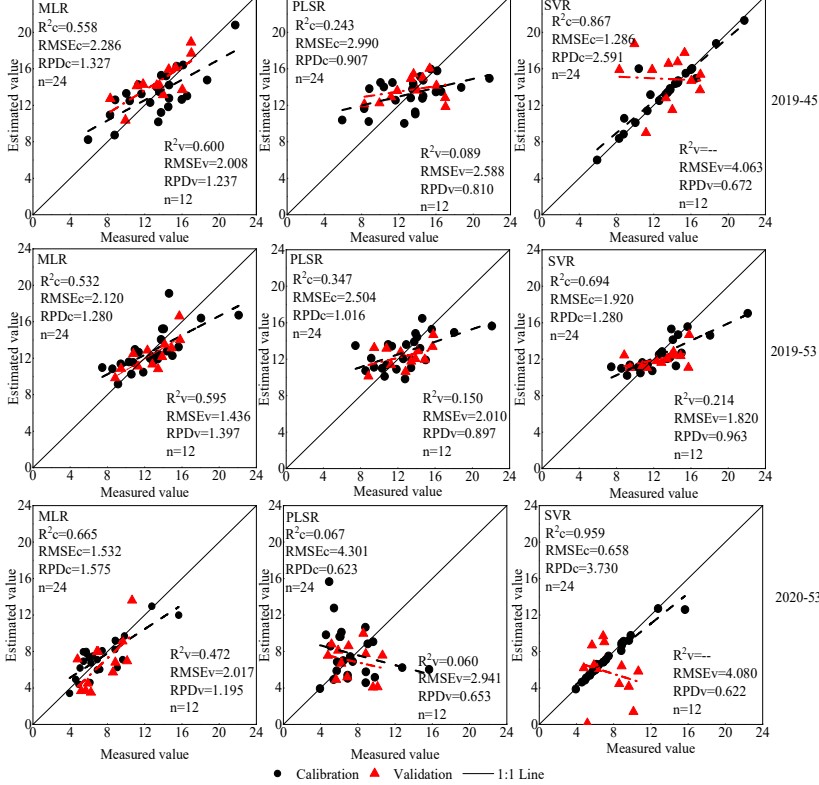

**Figure 11.** Parameters of CND for the estimation and verification of the model of drip-irrigation cotton at full-boll period based on vegetation index.

## 4. Discussion

This study explored the application of hyperspectral vegetation indexes in nitrogen estimation in cotton by examining the correlations between 30 published hyperspectral vegetation indexes and the LNC and CND of cotton. Some researchers believe that leaf nitrogen accumulation is affected by many factors, such as leaf nitrogen content, specific leaf weight, leaf area index, variety, and canopy structure, making it more difficult to estimate than the LNC. Therefore, the LNC estimation model is better than the leaf nitrogen accumulation estimation model [10,15]. The research of Xue et al. [41] showed that compared with the nitrogen content of rice leaves, the correlation between leaf nitrogen accumulation and canopy reflectance was consistent during the entire growth period, and the prediction ability of the leaf nitrogen accumulation estimation model was better. It has been considered that the effect of estimating the light layer index using original canopy spectral data is better than using the individual indexes. Zhao et al. [42] suggested that CND was a more sensitive parameter than LNC. The correlation coefficients between LNC, CND, and spectral index changed with the growth stage of winter wheat. A single spectral parameter was not dominant as the best variable, and the red-edge position was a good index for the estimation of winter wheat LNC. The same research showed that the red-edge chlorophyll index CIred-edge and REIPlinear were more sensitive to the canopy and nitrogen [10,11], consistent with the results of this study. Both LNC and CND screened indexes in this study included these two indexes. In addition, some studies have found that while there was no significant difference in leaf nitrogen content N% between different growth periods [10], the vegetation index would show large differences among nitrogen application levels and growth periods of crops [43]; this may also be the reason for the differences in cotton LNC and CND estimation based on hyperspectral vegetation indexes.

In this study, the nitrogen in each growth stage and in the entire growth stage of cotton was estimated by using the same selected group of hyperspectral vegetation indexes. The results showed that there was no extremely significant correlation between cotton leaf nitrogen concentration and canopy nitrogen density and each vegetation index at bud stage and initial flowering stage (Figures 5 and 6). There were weak correlations between the indexes of drip-irrigation cotton and vegetation index in each growth period. This study assumed that the use of hyperspectral vegetation indexes to estimate nitrogen during a single growth period is limited by the content difference. Previous studies had shown that a strong statistical relationship between spectral information and crop nitrogen status was not possible anywhere or at any time, suggesting that nitrogen inversion of a single spectral index should expand the sample size and coefficient of variation of samples and require more in-depth studies [19,44,45]. This result was also shown in the study of Li et al. [43] and Li et al. [46], where the correlation coefficients between 384 samples of winter wheat leaf nitrogen concentration and individual vegetation index were not ideal (r = 0.48 to 0.06). When using the entire growth period data and vegetation index input, there were strong correlations, indicating that the greater the sample difference, the better the correlation with vegetation index. The correlation between canopy nitrogen density and hyperspectral vegetation indexes increased in the full-boll period (Figure 6) and the $R^2$ of the model established by using multivariate vegetation index were higher in the flowering and boll period ($R^2_c$ = 0.544~0.658) and full-boll period ($R^2_c$ = 0.532~0.665). It may be that the planting density of drip-irrigation cotton in Xinjiang is high and reached the highest at full-boll period. However, this study used the hyperspectral vegetation indexes selected by correlation to estimate the same group of hyperspectral vegetation indexes, especially when the modeling method of MLR showed obvious modeling advantages. This demonstrated that the interaction between multiple hyperspectral vegetation indexes could improve the accuracy of the model and facilitate the application in agricultural production of remote sensing technology.

Existing research results have shown that the relationship between a single vegetation index and nitrogen in cotton is not a simple linear relationship. The relationship between the two is better fitted by a power function or exponential function, and the parameters of this

relationship are uncertain [43,44]. In this study, MLR, PLSR, and SVR modeling methods were used to estimate nitrogen during the entire growth period of drip-irrigation cotton. SVR showed strong advantages, consistent with the results of Yao et al. [20]. However, for the nitrogen estimation during individual growth periods, the model was prone to overfitting (Figures 7–10). For the estimation of nitrogen in cotton in the single growth periods, the research showed that the CND estimation model established by MLR was relatively stable and had strong applicability between years and varieties (Figures 8–11), especially for the later stages (flowering and boll stages and full-boll stage).

## 5. Conclusions

In this study, by analyzing the correlation between the canopy spectral index of two drip-irrigation cotton varieties and the main nitrogen indexes of cotton in 2019 and 2020, estimation models of cotton LNC and CND based on MLR, PLSR, and SVR were established. The main conclusions are as follows:

(1) The correlations between nitrogen indexes (LNC, CND) and 30 hyperspectral vegetation indexes in each growth stage of cotton were low in the early growth stages, and the growth stages with a strong correlation were in the late growth stage (i.e., flowering and boll stages and full-boll stage).

(2) TCARI, PRI, CCRI, SRI-2, and LNC had significant correlations between years and varieties. mND705, SRI-1, MSRI, MTCI, TCARI, NDVI705, VOG1, VOG2, VOG3, CCRI, CIred-edge, and REIPlinear had good and relatively stable correlation with cotton canopy nitrogen density between varieties and years. For the application of different modeling methods, when establishing the estimation models of cotton LNC and CND for the entire growth period, SVR showed a good modeling effect and could significantly improve the $R^2c$ of the model validation set, but the model established by SVR was prone to serious overfitting. For the establishment of a nitrogen in cotton estimation model for individual growth periods, SVR and PLSR were prone to overfitting, while the estimation model established by MLR had strong applicability between years and varieties.

(3) Based on multi-temporal nitrogen in cotton data and canopy spectral data, the modeling effect of canopy nitrogen density (population index) was better than leaf nitrogen concentration (individual index), and the estimation accuracy of the model in the later stages of cotton growth (flowering and boll stages and full-boll stage) was better than that in the early stages of cotton growth (bud stage and initial flowering stage).

**Author Contributions:** Conceptualization, L.M. and Y.M.; methodology, L.M. and X.C.; validation, Q.Z.and Q.Y.; formal analysis, X.C.; investigation, J.L.; resources, Z.Z. and X.L.; data curation, C.Y.; writing—original draft preparation, L.M. and X.C.; writing—review and editing, Q.Z.; visualization, X.C.; supervision, L.F.; project administration, Z.Z.; funding acquisition, Z.Z. and X.L. All authors have read and agreed to the published version of the manuscript.

**Funding:** The study was supported by National Natural Science Foundation of China, China (Grant No. 42061058), Science and Technology Research Plan for Key Areas of Xinjiang Production and Construction Corps, China (Grant No. 2020AB005) and Major Scientific and Technological Projects of Xinjiang Production and Construction Corps, China (Grant No. 2018AA00407). We thank Let-Pub (www.letpub.com, 22 October 2021) for its linguistic assistance during the preparation of this manuscript.

**Institutional Review Board Statement:** Not applicable.

**Informed Consent Statement:** Not applicable.

**Conflicts of Interest:** The authors declare that the research was conducted in the absence of any commercial or financial relationships that could be construed as a potential conflict of interest.

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
