# Peer review of "Estimation of Nitrogen Content Based on the Hyperspectral Vegetation Indexes of Interannual and Multi-Temporal in Cotton"

_agronomy, doi:10.3390/agronomy12061319_

Round 1

Reviewer 1 Report

Global comments:

The paper is about estimating cotton nitrogen, based on spectral indexes, not the other way around.

The paper is interesting, even if many of the correlations between the physical and spectral indexes are low. It will hep other researchers in their own work.

At some point you correlate nitrogen content with AGB, LAI and SLW. SLW is discarded because of its poor correlation with nitrogen content. But what do you do with AGB and LAI? What are they adding to the selection of hyperspectral indexes?

There are many comparaisons between physical indexes (LNC, CND, AGB) and spectral indexes, before selecting a few of the spectral indexes and building prediction models. This is a bit complex. Maybe you could add a diagram at the beginning of the paper to explain your methodology.

Discussion about which index best correlate with nitrogen at a given stage requires concentration to follow; maybe you could simplify this .

Since hyperspectral measurements are done from a distance of 1 meter, most of the field of view at the initial stages my consist in soil. How does this impact your measurements?

Some other comments:

Abstract

Line 32: ‘’ The estimation model of canopy nitrogen density was the most stable model established by MLR at the flowering and boll stages and the full boll stage’’; I would invert the sentence: “the canopy nitrogen density most stable model was established by MLR at the flowering and boll stages and the full boll stage with R2 =…” 

Introduction

Lines 64, to Line 76: some indices are common, others are less common. You should refer to table 3 for further explanation of what they are.

Line 76: I guess that R370 stand for the reflectance value at 370 nm. You should explain that.

Line 96 :” Xinjiang cotton needs 8~9 times the amount of water” to respect to what, other growing period?

Line 102: A vegetation index was selected and two employed in modeling. What is the first index selected for?

  1. Materials and Methods

Line 114 : Is the sunshine time over the 18 month period or an annual average?

Line 115: the accumulated temperature: Should the unit not be in degree-days or degree-hours?

Line 131: Others were managed… Other what? Others plots?

2.2  Data acquisition

2.2.1 what do the samples consist of? Hyperspectral measurements? Leaves?

2.2.4

FOV is 25° so at 1 meter above the plant, the FOV is around 45 cm. So at initial stage most of what is measured is soil?

Table 3: The PSRI and EVI indexes are listed twice with two different references

Line 183: End of title missing

2.4 Multivariate modeling method and model evaluation

I understand that you are defining training datasets and validation datasets as is done with deep learning models but less in conventional statistical analysis. Maybe you should specify how these two sets are defined, how many samples,..

3 Results

3.1

Table 4: RSD is not in % , but is the raw value between 0 and 1.

Line 203: LNC and CND for Xinluzao 53 were lower in 2020, despite a significant increase in the biomass. Indeed max value of LNC is higher in 2020, but mean value and mean + SD remain lower in 2020.

Line 220: not clear. I guess you want to say that LNC shows a poor correlation with the indexes for both varieties of cotton.

Line 222: correlation of CND with SLW is much lower than that with AGB or LAI.

Figure1: What the double stars  (**) stand for?

3.2

Lines 240-242: now vegetation indexes refer to hyperspectral vegetation indexes. You should mention this to ease the reading.

Figure2: some indexes such as CLA or NDVI705 used in this figure are not listed in Table 3

Figure 2: axes for LNC are reversed with positive values on the left. Is this correct? It looks strange that LNC is anticorrelated with EVI or DVI.

Figure 2: right end of the figure is out of view.

Figure2: different spans for the correlation coefficient horizontal axes make it difficult to compare the different flowering stages. This is also true for figure 3.

Line 256: sLAIDI seems to have a higher correlation coefficient than all the other indexes., including NDMI.

Lines 266 to 304 not easy to read. As for Table 5, having the indexes in alphabetical order would make the reading easier.

Lines 286 to 289 and Lines 301 to 304: these are the results of your analysis. These lines should be separated from the rest of the text to stand out from the mass of information in this chapter.

3.2.3

Line 305: Is this paragraph about LNC or LAI and AGB as indicated in Line 307 and Table 6?

Line 311to 314: you are considering that the correlation between NDVI for example and CND is at significant levels and stable despite correlation values not higher than 0.3. On line 321, on the contrary you are considering that the correlation between SRI2 and AGB is low despite values larger than 0.40 or 0.45. This is not consistent.

Line 328: the highest correlation value is – 0.533.

Line 330: “This may be due to the canopy spectrum adopted in this study.” What does this mean?

Line 350: what are you referring to ? Is thi for the training data set or the validation data set. I don’t see any 0.546 value in table 7.

3.3.2.

What is represented on figure 4? Is this R2 as we can guess from the text?

Some predictions are striking such as SVR for cotton bud stage. R2c is equal to 1 while R2v is 0. This is strange and requires further investigations even if we cannot expect fantastic predictions.

4 Discussion

Line 459: hyperspectral derived vegetation indexes

Line 463: references ?

Line 467: is this the R2 value from reference [7]?  Is the cotton variety in [7] comparable to the varieties in this paper?

 Line 493: I would add each selected vegetation index as some indexes showed good correlation EVI for the estimation of LNC at flowering stage.

Conclusions

In (2) I would add that you selected the indexes with the best correlation with LNC or CND

Line 536: ‘’ SVR showed a good modeling effect and could significantly improve the accuracy of the model” ; improve the accuracy with respect to what?

Author Response

We appreciate  anonymous referee for carefully reading the manuscript and for providing constructive comments. We believe that in dealing with all the comments, the clarity of the manuscript will be greatly improved.

Please see the attachment of detailed amendments.

Reviewer 2 Report

The article treats about modellig of nitrogen content with the use of spectral data. Most asumptions of the article are well-described, however several sections (regarding methodology) in the article must be double checked and corrected. Also, some minor English check must be performed.

Starting from the abstract, I would not use the phrase "cotton nitrogen" or "nitrogen cotton". It may be distracting for a reader. I would recommend using phrases like "the amount of nitrogen in cotton", "nitrogen in cotton", etc. This phrase also appear in line 42 and it must be corrected.
Also, in the abstract, if possible, please place absolute values of correlation. General information like "relatively stable" may be also misunderstood by the reader.

In lines 47 and 49 verbs "reflect" and "affect" should be used in 3-rd form as they refer to the nitrogen content.

In line 61 I would recommend using different citation regarding the NDVI index, as it refers to the water bodies, while the article treats about cotton.

In my opinion, the whole section from line 92 up to the end of introduction, should be placed in the "Methods and materials" section, because it refers to the specific parameters of experiment.

The amount of samples in "Data acquisition" section varies. It must be explained why the number of samples varies and later on in "Discussion" part it should be explained if it affects the model.

In section 2.2.2. should be explained if this method refers to some literature or if it is method developed by authors.

In 2.2.4 the name of the spectrometer manufacturer (Spectral Evolution) should be written with the use of capital letters or even cited.

Section 2.4. - mistake in the section title

Generally, in section 2, there should also be an explanation how the specific indices were calculated. Resolution of spectrometer does not fit the wavelenghts in some indices. Was some bandwith range used for the index calculation? Please provide set of spetral curves on the graph for some representative categories and/or raw data from spectrometers.

To the later parts I do not have much comments. The results are generally well described, so as discussion and results part.

Of course, graphical visualisation of the experiments and tables may be corrected and improved:
-improve colorset in Figure 1.
-labeling fields at Fig. 2 and 3.
-making consistent decimal numbers on table 5 and 6

Author Response

(The authors gave the same response as above.)

Round 2

Reviewer 1 Report

Significant editing of the initial version of the paper was made to address reviewers’ comments. I think that this improves the readability of the paper.

Leaf Nitrogen Content prediction is not so good. Only Canopy Nitrogen content is fair to good at the late stages. Even if it would be better for the authors to show strong predictions, showing that predictions are not very good is interesting for the scientific community.

Some comments on this revised version:

Line 89: you seem to oppose deep learning methods with using spectral features or indexes. Deep learning or machine learning methods, such as the ones mentioned in the following lines of your paper, can be applied on raw reflectance or on spectral indexes or alternatively after spectral dimension reduction.

Line 102: sorry, I still do not understand; Xinjian cotton needs 8-9 times the amount of water, compared to what?

Line 109: indexes or indices instead of indexs

Line 119: Lai instead of LAI

Line 145: the planting density value is surprising, at least the exponent is certainly wrong.

Line 210 and line 211: paragraph title modified but still incomplete

Line 260: regarding SLW this sentence is not consistent with what you write on line 270 and 271 and what is illustrated on Figure 4.. I would remove SLW from line 260.

Line 264: Figure 1 is now Figure 4.

Lines 352-354: correlation with CND or LAI? In addition, correlation with indexes  such as NDVI, SRI-1 are not really significant or stable

Line 359: which vegetation index are you referring to?

Line 375: TCARI also has good correlation with LAI and AGB.  Globally variety 2020-53 has lower to much lower correlations with hyperspectral indexes compared to 2019-45 variety. Does this have an explanation?

Line 382: analysis in 3.2.2 (and 3.2.3 ) instead of 2.2.2; I would suggest to recall which are the indexes that will be used.

Line 399: where is this value of .232 coming from. I don’t see it in table 7.

Predictions on the validation dataset for LNC for variety 2019-45 are not good; while predictions are better for CND and this variety, in line with good correlation coefficients in 3.2.3. It is the opposite for variety 2020-53 : good predictions for LNC , poor for CND. Should we conclude that there is no model suitable for all the varieties for any of the LNC and CND indexes? Models have to be adapted for each variety and indexes.

Line 492: prediction of LNC is poor at any of the stages with all the methods. Prediction of CND is indeed better particularly at the later stages.

I am surprised by some of your data: Figure 10, PLSR, 2019-53 : R2v is 0.774 where it seems very close to zero.

Discussion

Line 503: nitrogen in cotton indexes.

Line 562: especially for the later stages (flowering and boll stages and full-boll stage) since MLR predictions are quite poor at initial bud stage both for 2019-45 and 2020-53.

Line 571: with a correlation coefficient up to 0.6.

Line 572: I would not say ‘very significant’ for correlations that do not reach 0.4 in the best case for TCARI and CCRI, and barely exceed .4 for PRI and one of the variety-year. ‘significant’ would be more appropriate.

Author Response

Response to Reviewer 1 Comments

Comments and Suggestions for Authors

General comment:

Significant editing of the initial version of the paper was made to address reviewers’ comments. I think that this improves the readability of the paper.

Leaf Nitrogen Content prediction is not so good. Only Canopy Nitrogen content is fair to good at the late stages. Even if it would be better for the authors to show strong predictions, showing that predictions are not very good is interesting for the scientific community.

Response : We thank the reviewer for their affirmation of us, in terms of the modification of the paper and the reply to the reviewers. Below are our responses (in red color) to their comments resulting in a number of clarifications. This may be our future research content of us.

Some comments on this revised version:

Comment 1:

Line 89: you seem to oppose deep learning methods with using spectral features or indexes. Deep learning or machine learning methods, such as the ones mentioned in the following lines of your paper, can be applied on raw reflectance or on spectral indexes or alternatively after spectral dimension reduction. 

Response 1: We are very sorry to give this conclusion to the reviewer. We have no objection deep learning methods with using spectral features or indexes. Just based on the results of our research, multiple linear regression is more suitable for our research. Limited to our sample size, PLSR and SVR do not reflect their advantages. We did not try the impact of spectral data processing methods on our research, such as the calculation of vegetation index using first derivative spectrum.

Comment 2:

 Line 102: sorry, I still do not understand; Xinjiang cotton needs 8-9 times the amount of water, compared to what?

Response 2: We are very sorry for this confusion. “8-9 times” represents the frequency of fertilizer application. According to the actual situation of this study, we modified "8~9 times" to "8 times"in line 102. Cotton planting in Xinjiang adopts "water and fertilizer integration technology". We dissolve the applied fertilizer in water and use drip irrigation belt to send the water with fertilizer to the root of cotton. From the emergence of cotton seedlings to the harvest, we have drip irrigation for 11 times, of which the water containing fertilizer is 8 applications. In order to make our explanation clearer, we list the cotton drip irrigation dates and proportions in 2019 and 2020 in Table 1(Line 159).

Table 1. Application amount of integrated drip irrigation of water and fertilizer in field experiment

Date(2019)

Fertilizer percent

Water volume (m3/m2)

Date (2020)

Fertilizer percent

Water volume (m3/m2)

4-29

0%

0.022

4-30

0%

0.022

5-02

0%

0.030

5-05

0%

0.030

6-14

5%

0.033

6-15

5%

0.033

6-22

10%

0.060

6-24

10%

0.061

6-30

15%

0.051

7-05

15%

0.051

7-09

20%

0.045

7-14

20%

0.043

7.18

25%

0.049

7.20

25%

0.049

7.25

12%

0.045

7.27

12%

0.045

8.03

8%

0.042

8.02

8%

0.042

8.12

5%

0.034

8.12 

5%

0.034

8.18

0%

0.037

8.19

0%

0.037

Comment 3:

Line 109: indexes or indices instead of indexs

Response 3: We are sorry for this mistake. We have revised "indexes" to "indexs" in line109.  

Comment 4:

Line 119: Lai instead of LAI

Response 4: We are sorry for this mistake. We have revised "indexes" to "indexs" in line 119

Comment 5:

Line 145: the planting density value is surprising, at least the exponent is certainly wrong.

Response 5: We sincerely acknowledge the referee for her/his careful review of the manuscript abd we are sorry for this mistake. We have revised "21.50×10-4 plants/ha" to "21.50×104 " in line 147. About the planting density of cotton in Xinjiang, we explain as follows: In order to effectively use light and heat resources, particularly to avoid the adverse effects of limited growing season, a high-yielding cultivation pattern called “short-dense-early” has been widely adopted in Xinjiang cotton region. "Dense" represents the high-density planting mode of cotton in Xinjiang. The planting density is always in the range of 200,000–300,000 plants/ha.

  • Dai J , Dong H. Intensive cotton farming technologies in China: Achievements, challenges and countermeasures[J]. Field Crops Research, 2014, 155:99-110.
  • Feng L , Dai J ,  Tian L , et al. Review of the technology for high-yielding and efficient cotton cultivation in the northwest inland cotton-growing region of China[J]. Field Crops Research, 2017, 208:18-26.

Comment 6:

Line 210 and line 211: paragraph title modified but still incomplete

Response 6: We are sorry for this mistake. We have revised "Modeling method and evaluation method" to "Model establishment method and model evaluation index" in line 225.

Comment 7:

Line 260: regarding SLW this sentence is not consistent with what you write on line 270 and 271 and what is illustrated on Figure 4.. I would remove SLW from line 260.

Response 7: We agree with the referee. We have removed SLW from line 274.

Comment 8:

Line 264: Figure 1 is now Figure 4.

Response 8: We are sorry for this mistake. we modified "Figure 1" to " Figure 4"in line 278.

Comment 9:

Lines 352-354: correlation with CND or LAI? In addition, correlation with indexes  such as NDVI, SRI-1 are not really significant or stable

Response 9: We have revised “the correlation coefficient with Xinluzao 53 was -0.422 , but the correlation coefficient with Xinluzao 45 was only 0.131. ” to “The correlation coefficient between DVI with LAI was -0.422 in Xinluzao 53 of 2019, but the correlation coefficient in Xinluzao 45 of 2019 was only 0.131. ” in line 366~368. 

The correlation coefficient between NDVI, SRI-1 and LAI was 0.282, 0.163, 0.234 and 0.186, 0.163, 0.211 between years and varieties(Tabel 7) , the correlation was positive, and reached a significant level at the level of P < 0.05, so we believed that there was a stable and significant correlation.

We modified "The correlations between NDVI, SIPI, mND705, SRI-1, MSRI, MTCI, MCARI, TCARI, NDVI705, VOG1, VOG2, VOG3, PRI, SRI-2, CIred-edge, and REIPlinear (16 indexes) and CND reached significant levels between years and varieties, and the correlations were relatively stable" to "The correlations between NDVI, SIPI, mND705, SRI-1, MSRI, MTCI, MCARI, TCARI, NDVI705, VOG1, VOG2, VOG3, PRI, SRI-2, CIred-edge, and REIPlinear (16 indexes) and LAI reached significant levels (p<0.05)between years and varieties, and the correlations were relatively stable” in line 368~371.

Comment 10:

Line 359: which vegetation index are you referring to?

Response 10: We are sorry for this inexact wording. We have revised “ Compared with the correlation between LAI and hyperspectral vegetation index” to  “Compared with the correlation between LAI and hyperspectral vegetation indexes of the band with red edge, such as MSRI, MTCI, VOG1, VOG2, VOG3, CIred-edge, and REIPlinear,” in line 374~376.

Comment 11:

Line 375: TCARI also has good correlation with LAI and AGB.  Globally variety 2020-53 has lower to much lower correlations with hyperspectral indexes compared to 2019-45 variety. Does this have an explanation?

Response 11: According to the research results of other researchers, different varieties of the same crop will cause differences in hyperspectral reflectance. This difference may be caused by crop plant height, canopy coverage , climatic conditions and other factors. The two cotton varieties in this study have differences in leaf color and plant type(Xinluzao 45 (type II fruit branch, light green of leaf color ) and Xinluzao 53 (type I fruit branch, dark green of leaf color), Figure 2, in line 143~144 ), which may lead to the difference of vegetation index between the two varieties. However, we have not found any specific research in this area, and this paper does not involve detailed research in this area. We think this is a good research direction. In the future research, we will deeply study the influence mechanism of cotton leaf color and plant type on Hyperspectral.

[1] Flores M , Paschoalete W M , Baio F , et al. Relationship between vegetation indices and agronomic performance of maize varieties under different nitrogen rates[J]. Bioscience Journal, 2020, 36(5).

[2] Tong Q , Zhao Y , Xia Z , et al. New progress in study on vegetation models for hyperspectral remote sensing[J]. Proceedings of SPIE - The International Society for Optical Engineering, 2001, 4151:143-152.

Figure 2. The top view of cotton test area(Left)

Comment 12:

Line 382: analysis in 3.2.2 (and 3.2.3 ) instead of 2.2.2; I would suggest to recall which are the indexes that will be used.

Response 12: We are sorry for this mistake. We have revised “ 3.2.2” to “2.2.2” in line 401and added the indexes that will be used (The model parameters of LNC were TCARI, PRI, CCRI, SRI-2, and the model parameters of CND were MTCI, NDVI705, VOG1, VOG2, VOG3, CIred-edge, and REIPlinear) in line403~405.

Comment 13:

Line 399: where is this value of .232 coming from. I don’t see it in table 7.

Predictions on the validation dataset for LNC for variety 2019-45 are not good; while predictions are better for CND and this variety, in line with good correlation coefficients in 3.2.3. It is the opposite for variety 2020-53 : good predictions for LNC , poor for CND. Should we conclude that there is no model suitable for all the varieties for any of the LNC and CND indexes? Models have to be adapted for each variety and indexes.

Response 13: We are sorry for this mistake. We have revised “0.232” to “0.113” in line 420. I agree with the reviewer's conclusion to some extent. Because there are many factors affecting hyperspectral, the varieties adopted by postgraduates and the climatic conditions of the experimental area all have differences on the research results. I think this is also the main reason why researchers use different modeling methods and data processing methods to estimate nitrogen in different crops and different ecological areas. This is also the reason why this study uses different years and different varieties to monitor cotton multi temporal nitrogen.

Comment 14:

Line 492: prediction of LNC is poor at any of the stages with all the methods. Prediction of CND is indeed better particularly at the later stages.

I am surprised by some of your data: Figure 10, PLSR, 2019-53 : R2v is 0.774 where it seems very close to zero.

Response 14: PLSR belongs to multiple linear regression, which will lead to over fitting due to the correlation between independent variables, and the amount of data and the coefficient of variation of samples can lead to over fitting of the model. We analyzed this aspect in the discussion part in line 553~557.

Discussion

Comment 15:

Line 503: nitrogen in cotton indexes.

Response 15: We modified this sentence, revised “This study explored the application of hyperspectral vegetation indexes in nitrogen in cotton estimation by examining the correlations between 30 published hyperspectral vegetation indexes and the nitrogen in cotton indexes.” to “This study explored the application of hyperspectral vegetation indexes in nitrogen estimation in cotton by examining the correlations between 30 published hyperspectral vegetation indexes and the LNC and CND in cotton.” in line 522~524.

Comment 16:

Line 562: especially for the later stages (flowering and boll stages and full-boll stage) since MLR predictions are quite poor at initial bud stage both for 2019-45 and 2020-53.

Response 16: We appreciate the referee for carefully reading the manuscript and supplying us with useful suggestions. we modified "the research showed that the CND estimation model established by MLR was relatively stable and had strong applicability between years and varieties (Figure 8-Figure 11)" to "the research showed that the CND estimation model established by MLR was relatively stable and had strong applicability between years and varieties (Figure 8-Figure 11), especially for the later stages (flowering and boll stages and full-boll stage)"in line 583~584. 

Comment 17:

Line 571: with a correlation coefficient up to 0.6.

Response 17: we are very sorry for the excessive wording. We removed “with a correlation coefficient up to 0.6.” in line 593.  

Comment 18:

Line 572: I would not say ‘very significant’ for correlations that do not reach 0.4 in the best case for TCARI and CCRI, and barely exceed .4 for PRI and one of the variety-year. ‘significant’ would be more appropriate.

 Response 18: We thank the reviewer for the useful suggestion and we followed this suggestion. We have revised “very significant” to “significant” in line 594.

Reviewer 2 Report

Thank you for your reply. Most of my comments were taken into account.

The article has been significantly improved by making the right corrections to all reviews. 

However, there are still some minor errors in the text and it should be double checked before publication. Also, some figer may be improved (ex. Figure 4 has still illegible labels and colors can be improved).

Author Response

Response to Reviewer 2 Comments

General comment:

Thank you for your reply. Most of my comments were taken into account.

The article has been significantly improved by making the right corrections to all reviews. 

Response :  We thank the reviewer for their affirmation of us, in terms of the modification of the paper and the reply to the reviewers. 

Comment 1:

However, there are still some minor errors in the text and it should be double checked before publication. Also, some figer may be improved (ex. Figure 4 has still illegible labels and colors can be improved).

Response 1: We have modified the color of Figure 4 through the discussion on the color matching of the chart in the paper, and please review the manuscript.

Figure 4. Correlations between nitrogen and AGB, LAI, and SLW at different growth stages of cotton in 2019 (a: Xinluzao 45, b: Xinluzao 53)(* indicates that significant difference is achieved under P < 0.05; ** indicates that significant difference is achieved under P < 0.01 )
